# CUT&Tag recovers up to half of ENCODE ChIP-seq histone acetylation peaks

Leyla Abbasova [1,2,5], Paulina Urbanaviciute[3,4,5], Di Hu[1,2,5], Joy N. Ismail[3,4], Brian M. Schilder[1,2], Alexi Nott [1,2], Nathan G. Skene[1,2] & Sarah J. Marzi [2,3,4] ✉

DNA-protein interactions have traditionally been profiled via chromatin immunoprecipitation followed by next-generation sequencing (ChIP-seq). Cleavage Under Targets & Tagmentation (CUT&Tag) is a rapidly expanding technique that enables the profiling of such interactions in situ at high sensitivity. However, thorough evaluation and benchmarking against established ChIP-seq datasets are lacking. Here, we comprehensively benchmarked CUT&Tag for H3K27ac and H3K27me3 against published ChIP-seq profiles from ENCODE in K562 cells. Combining multiple new and published CUT&Tag datasets, there was an average recall of 54% known ENCODE peaks for both histone modifications. We tested peak callers MACS2 and SEACR and identified optimal peak calling parameters. Overall, peaks identified by CUT&Tag represent the strongest ENCODE peaks and show the same functional and biological enrichments as ChIP-seq peaks identified by ENCODE. Our workflow systematically evaluates the merits of methodological adjustments, providing a benchmarking framework for the experimental design and analysis of CUT&Tag studies.

In recent years, the field of epigenetics has attracted increasing interest as a source of new insights into the mechanisms underlying human disease. Human disease risk variants identified through genome-wide association studies (GWAS) overwhelmingly localize to non-coding regions of the genome[1–4]. These risk variants appear to be enriched in gene regulatory regions[5–7]. Chromatin dynamics at regulatory regions are governed by nucleosomes and their post-translational modifications, as well as interacting chromatin-associated complexes and transcription factors. Chromatin marks can define regions of activation and silencing and mark transcriptional regulatory elements. These can be cell type-specific and are known to be dynamic throughout development, aging, and disease progression[8]. Disease risk variants appear to be specifically enriched in active regulatory elements, particularly those marked by H3K27ac[6,7]. H3K27ac is a highly cell type-specific histone modification and a marker of active enhancers and promoters[9], which has been implicated in complex diseases. For

example, in the brain, variation in H3K27ac has been associated with neurodegenerative and neuropsychiatric disorders, including Alzheimer's disease[6,10,11]. However, understanding the precise regulatory mechanisms underlying epigenetic regulation in complex human disease and linking non-coding variants to disease phenotypes has been impeded by a lack of epigenomic annotations in disease and control tissues. Furthermore, the resources that do exist tend to use bulk tissues of heterogeneous organs, which are characterized by epigenomic signatures that are predominantly influenced by cell type composition and obscure cell type-specific regulatory landscapes.

For many years, chromatin immunoprecipitation followed by next-generation sequencing (ChIP-seq) has served as a standard method for epigenomic profiling. In ChIP-seq, chromatin is first cross-linked and solubilized, after which a primary antibody specific for the histone mark of interest enables immunoprecipitation of bound DNA[12]. However, it has potential limitations, such as low signal-to-noise ratio,

---

[1]UK Dementia Research Institute at Imperial College London, London, UK. [2]Department of Brain Sciences, Imperial College London, London, UK. [3]UK Dementia Research Institute at King's College London, London, UK. [4]Department of Basic and Clinical Neuroscience, Institute of Psychiatry, Psychology and Neuroscience, King's College London, London, UK. [5]These authors contributed equally: Leyla Abbasova, Paulina Urbanaviciute, Di Hu. ✉e-mail: sarah.marzi@kcl.ac.uk

epitope masking from fixation and cross-linking, and heterochromatin bias from chromatin sonication[13,14]. ChIP-seq poses challenges when working with low cell numbers, requiring approximately 1-10 million cells as input, with high demands on sequencing coverage, due to the low signal-to-noise ratio. In addition, ChIP-seq does not adapt well to single-cell applications due to its high cell input requirements and poor signal specificity. Cleavage Under Targets & Tagmentation (CUT&Tag) is an enzyme-tethering approach that has been presented as a streamlined, easily scalable, and cost-effective alternative to ChIP-seq. CUT&Tag has been reported to have superior chromatin mapping capabilities as compared to ChIP-seq at approximately 200-fold reduced cellular input and 10-fold reduced sequencing depth requirements[15]. CUT&Tag uses permeabilized nuclei to allow antibodies to bind chromatin-associated factors, which enables the tethering of protein A-Tn5 transposase fusion protein (pA-Tn5). Upon activation of pA-Tn5, cleavage of intact DNA and insertion of adapters (tagmentation) occurs for paired-end DNA sequencing. Following tagmentation, DNA fragments remain inside the nucleus, making the method amenable to single-cell chromatin profiling applications, for example, enabling individual sorting of nuclei and PCR barcoding. The increased signal-to-noise ratio of CUT&Tag for histone marks is attributed to the direct antibody tethering of pA-Tn5 and its integration of adapters in situ while it stays bound to the antibody target of

interest during incubation. The process involves minimal sample loss with direct enzymatic end-polishing and ligation compared to regular library preparation protocols that result in sample loss, including ChIP-seq and CUT&RUN[15].

For ChIP-seq, experimental and analytical guidelines as well as datasets generated by the Encyclopedia of DNA Elements (ENCODE) consortium, have served as standard references in the field for years[16]. In contrast, as a relatively new method, CUT&Tag lacks equivalent systematic optimization or benchmarking against existing datasets, and there is no established consensus regarding experimental recommendations and data analysis workflows. Here, we undertook experimental optimizations and systematic benchmarking of CUT&Tag against ENCODE in human K562 cells for histone modifications H3K27ac and H3K27me3 to serve as a guide for the design and analysis of future CUT&Tag studies. Since the development of CUT&Tag has primarily assessed methyl marks, where H3K27me3 is the recommended positive control[17], we focused in-depth on underexplored H3K27ac, testing multiple ChIP-grade antibody sources[6,10,18,19], antibody dilutions, histone deacetylase inhibitors (HDACi), as well as PCR parameters, and DNA extraction methods for library preparation (Fig. 1a). Experimental outcomes were evaluated by quantitative polymerase chain reaction (qPCR) and paired-end genomic sequencing. Our computational workflow served to iteratively guide

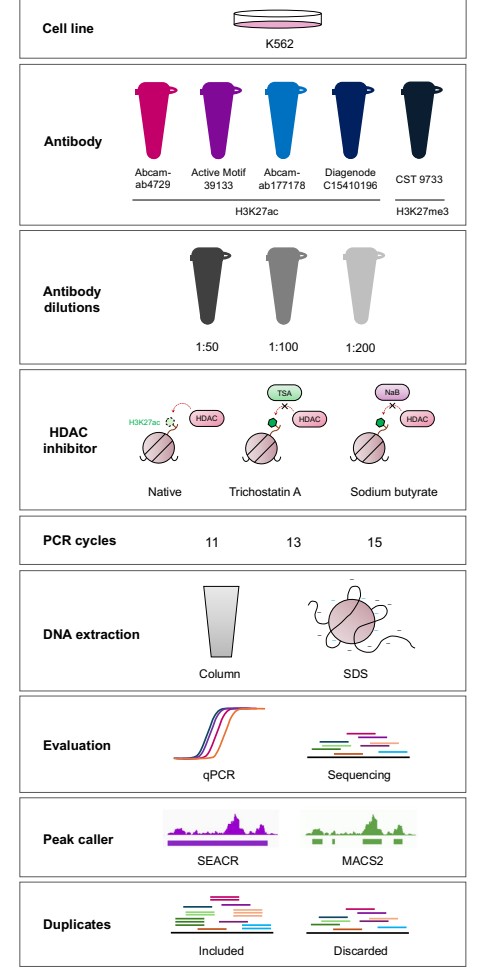

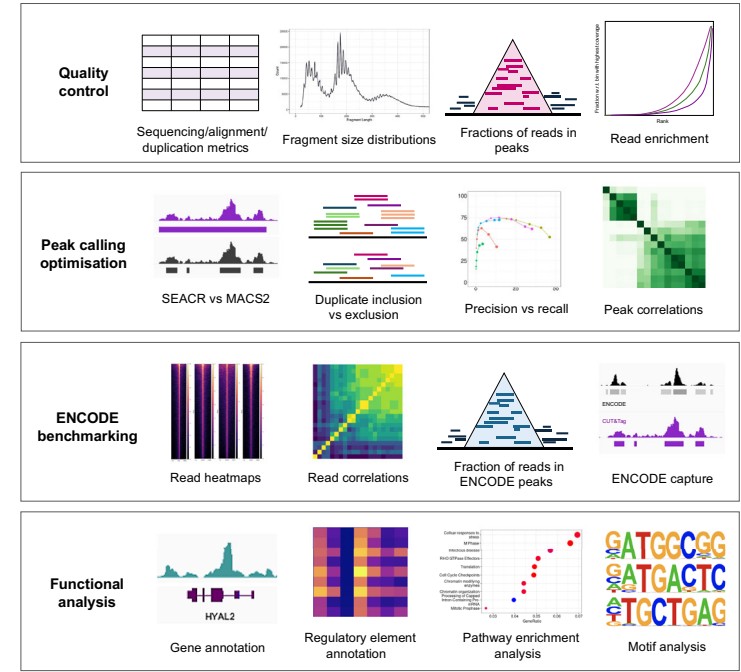

**Fig. 1 | Overview of experimental design and computational benchmarking.** **a** Summary of experimental design. Five antibodies were tested at dilutions 1:50, 1:100, and 1:200, and 11, 13, or 15 PCR cycles for library preparation. H3K27ac libraries were assessed with and without HDAC inhibitor Trichostatin A (TSA; 1 μM) or sodium butyrate (NaB; 5 mM). Column- and SDS-based DNA extraction methods were compared. Antibody performance was assessed by qPCR and sequencing, and

sequenced reads were processed with and without duplicates using peak callers SEACR and MACS2. **b** Summary of analytical approaches. Analysis comprised quality control of sequencing data, optimization of peak calling approaches with both peak callers, and comparison between CUT&Tag and ENCODE datasets at the level of reads, peaks, and functional annotation.

experimental optimizations, appraise CUT&Tag data quality, and benchmark CUT&Tag performance against ENCODE ChIP-seq profiles (Fig. 1b). We explored the suitability of different peak calling approaches (SEACR and MACS2), and the effects of inclusion versus exclusion of PCR duplicate reads. We characterized the similarities and differences between CUT&Tag and ENCODE ChIP-seq based on parameters including read- and peak-level correlation, regulatory element annotation, gene ontology enrichment, and transcription factor binding motif (TFBM) analysis. We developed a benchmarking pipeline, EpiCompare[20], to help researchers optimally analyze and interpret CUT&Tag studies.

## Results

### Overview of systematic H3K27 CUT&Tag experimental design and analysis

To benchmark the performance of CUT&Tag against established ENCODE ChIP-seq, we profiled histone modifications H3K27ac, a marker of active enhancers and promoters, and H3K27me3, associated with heterochromatin and assessed in the original series of papers introducing CUT&Tag[15,17,21]. Both histone modifications were characterized in K562 cells, generating a total of 38 new CUT&Tag sequencing datasets. We undertook systematic experimental optimizations for H3K27ac CUT&Tag testing multiple ChIP-grade antibody sources[6,10,18,19], antibody dilutions (1:50, 1:100, 1:200), as well as different PCR cycle numbers, DNA extraction methods for library preparation, and histone deacetylase inhibitors (HDACi; Fig. 1a). Primary conditions were first validated by performing qPCR using positive and negative control primers designed based on ENCODE ChIP-seq peaks (Supplementary Table 1). The best conditions were subsequently subjected to paired-end sequencing. Our computational workflow iteratively guided experimental optimizations, assessed data quality, and benchmarked CUT&Tag performance against ENCODE ChIP-seq (Fig. 1b).

### Experimental optimization of CUT&Tag

We first assessed four ChIP-seq grade H3K27ac antibodies across three dilutions (1:50, 1:100 and 1:200) by qPCR, using primers designed to amplify regions corresponding to genes falling into the most significant ENCODE peaks (positive controls: *ARGHAP22*, *COX4I2*, *MTHFR*, *ZMYND8*) versus least significant ENCODE peaks (negative controls: *KLHL11*, *SIGIRR*) (Methods; Supplementary Table 1; Fig. 2a). Based on the outcome, we selected Abcam-ab4729 (1:100; the same antibody was used in ENCODE ChIP-seq), Diagenode C15410196 (1:50 and 1:100), Abcam-ab177178 (1:100), and Active Motif 39133 (1:100) for sequencing. These antibodies will be henceforth referred to as Abcam-ab4729 (ab4729), Diagenode (diag), Abcam-ab177178 (ab177), and Active Motif. H3K27me3 CUT&Tag was profiled using ChIP-grade antibody Cell Signaling Technology-9733, the same antibody used in ENCODE, at a dilution of 1:100 as previously recommended[17]. In-house samples were compared with published CUT&Tag[17] and CUT&RUN[22] data from the research group that originally developed these methods.

Since H3K27ac is dynamically deposited and removed by histone acetyltransferases and deacetylases (HDACs), chromatin mapping methods can potentially benefit from adding HDACi to eliminate residual deacetylase activity and thereby stabilize acetyl marks. This is particularly relevant for CUT&Tag, which is carried out under native conditions where residual HDAC activity may have a greater impact. To test whether the addition of a potent HDAC inhibitor improves data quality and ENCODE coverage of previously tested antibodies, H3K27ac CUT&Tag was performed with the addition of Trichostatin A (TSA; 1 μM). This data was compared to original samples scaled to the same read depths. Addition of TSA did not consistently increase total peak detection using MACS2 (q-value threshold $1\times10^{-5}$, nolambda, nomodel) or SEACR (stringent settings and threshold 0.01 (Supplementary Fig. 1a) and did not improve signal to noise ratio

(Supplementary Fig. 1b) or ENCODE capture (Fig. 2b). Here, ENCODE capture was assessed using two metrics: precision (the proportion of CUT&Tag peaks falling into ENCODE peaks of the same histone modification) and recall (the proportion of ENCODE peaks captured by CUT&Tag). H3K27ac CUT&Tag was also attempted with the addition of sodium butyrate (NaB; 5 mM), and libraries were evaluated by qPCR, which revealed no improvement in CUT&Tag binding signal (Supplementary Fig. 1c).

Preliminary analysis of sequencing data revealed high duplication rates across all samples (min: 55.49%; max: 98.45%; mean: 82.25%; Supplementary Table 2). CUT&Tag library preparation was initially carried out with 15 PCR cycles, as per the original protocol[16]. To test whether this contributed to high numbers of duplicate reads, we carried out CUT&Tag library preparation at 11 and 13 PCR cycles. In addition to varying cycle numbers, we also tested SDS-based versus column-based methods of DNA extraction (see Methods). All samples were analyzed at the original read depth (Fig. 2c–e) and down sampled to the shared minimum read depth (2.6 million paired-end reads; Supplementary Fig. 1d–f) to compare duplication rates, total unique fragments, and ENCODE coverage. Varying PCR cycles while employing SDS-based DNA extraction produced mixed changes in duplication rate, whereas samples obtained with column-based extraction showed an increase in duplication rate from 11 to 13 PCR cycles (Fig. 2c). Overall, the greatest numbers of unique fragments were generated using 15 PCR cycles and SDS-based DNA extraction (Fig. 2d), although the difference was less significant after down sampling (Supplementary Fig. 1e). Almost all samples captured ENCODE peaks with high precision regardless of condition and analysis approach, but total ENCODE recall by Abcam-ab4729 and Diagenode (1:50) was improved when using 15 PCR cycles (Fig. 2e). The superior unique fragment yield at 15 PCR cycles did not translate into improved ENCODE coverage after down sampling (Supplementary Fig. 1f). Based on these optimizations, the 15 PCR cycle, SDS-based DNA extraction experiments without addition of HDACi were taken forward for systematic benchmarking.

### Quality control of CUT&Tag data

To ensure robust quality control and analytical benchmarking, we generated two additional sequencing datasets with lower duplication rates for the best performing antibodies: Abcam-ab4729, Abcam-ab177178, and Diagenode for H3K27ac, and CST-9733 for H3K27me3 (Supplementary Table 3). We first quantified fragment length and observed fragment sizes comparable to CUT&Tag in human nuclei, with an abundance of fragments at around 180 bp in size, reflecting the length of DNA from a single nucleosome (Fig. 3a; Supplementary Fig. 2a)[23,24]. We also observed short fragments (<100 bp) similar to previous CUT&Tag data[17], potentially caused by tagmentation of open chromatin[25]. Shorter fragments were not more abundant in duplicate-containing samples, suggesting that these are not a consequence of PCR amplification bias[26].

We next evaluated signal-to-noise quality by calculating the fractions of CUT&Tag reads in peaks (FRiPs) defined in our dataset, as well as pre-defined ENCODE peaks. Specifically, we compared our data with ENCODE H3K27ac narrow and H3K27me3 broad peak sets (Fig. 3b). To identify peaks in our CUT&Tag samples, we used two analytical approaches: (1) MACS2, a standard peak caller for ChIP-seq data used by ENCODE that was also applied to recent CUT&Tag datasets, and (2) SEACR, an algorithm developed specifically to detect peaks in high signal-to-noise data, such as CUT&RUN and CUT&Tag[27,28].

FRiP scores for H3K27ac CUT&Tag sample peaks were comparable across antibodies and peak callers (ab-177, ab-4729, diag, MACS2 mean = 38.23, 32.22, 42.87, sd = 0.44, 12.05, 4.82, respectively; SEACR mean = 40.78, 33.86, 43.07, sd = 0.10, 6.69, 4.16, respectively). These were also highly similar to FRiP scores in pre-defined ENCODE peaks (mean = 37.16, sd = 6.29) and close to the reported ENCODE ChIP FRiP

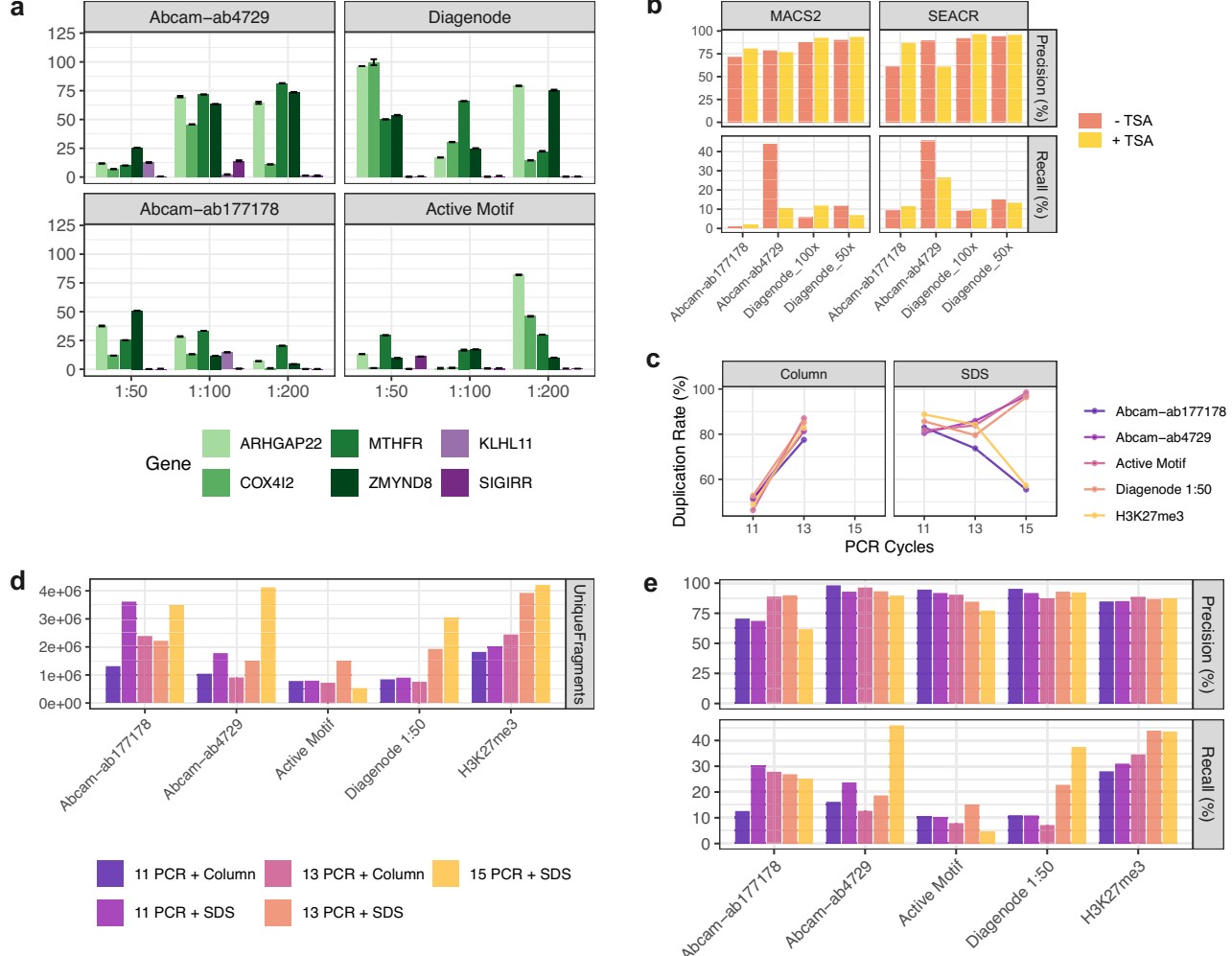

**Fig. 2 | Experimental optimization of CUT&Tag. a** Results of qPCR amplification of genes falling into most significant ENCODE H3K27ac peak regions in CUT&Tag samples. Data are shown as ΔΔCt for triplicate replicates, and error bars represent ΔΔCt ± SD. **b** Precision and recall of ENCODE H3K27ac peak capture with and without the addition of HDAC inhibitor TSA. Reads from each antibody were down sampled to match the condition (with or without TSA) with the lower read depth. **c** Duplication rates (% of mapped reads) across different antibodies and experimental conditions. **d** Total unique fragments obtained across experiments. **e** ENCODE capture metrics obtained using column- or SDS-based DNA extraction and 11, 13, or 15 PCR cycles for sequencing library preparation at the original read depth and SEACR peak calling.

score of 42% (Fig. 3b; Supplementary Fig. 2b). Removing duplicate reads yielded slightly higher Diagenode MACS2, SEACR, and ENCODE FRiP scores, although for the remaining antibodies this effect was not observed. We did find that for some samples with high duplication rates, FRiP scores for MACS2 peaks were inflated (Supplementary Fig. 2b). Of the H3K27ac antibodies tested, Diagenode, at 1:50 dilution without duplicates, showed the highest percentage of reads falling into published ENCODE H3K27ac peaks (mean = 43.43, sd = 2.56). This was closely followed by one of the Abcam-ab4729 samples (ab-4729-2; 39.28). For H3K27me3, CUT&Tag sample FRiPs (MACS2 mean = 73.65, sd = 1.11; SEACR mean = 72.43, sd = 2.74) outperformed the ENCODE reported H3K27me3 FRiP score of 66%, although these were significantly lower than the CUT&Tag FRiPs in ENCODE H3K27me3 regions (mean = 85.80, sd = 0.81) (Fig. 3b).

We quantified the specificity of CUT&Tag reads in ENCODE peaks of the corresponding histone modification as a proportion of CUT&Tag reads in ENCODE peaks of the other modification. Of note, H3K27me3 CUT&Tag reads show a highly specific enrichment at ENCODE H3K27me3 peaks, while H3K27ac CUT&Tag produces more residual reads aligning to some ENCODE H3K27me3 locations, both for

in-house (H3K27me3: mean = 0.99, sd = 1.74 × 10⁻³; H3K27ac: mean = 0.65, sd = 6.54 × 10⁻²) and published data (H3K27me3: mean = 0.99, sd = 2.12 × 10⁻³; H3K27ac: mean = 0.66, sd = 1.23 × 10⁻³). Among the tested H3K27ac antibodies, the highest ENCODE enrichment was seen with the Diagenode antibody (mean = 0.72, s d = 2.23 × 10⁻²). Visualization in the Integrative Genomics Viewer (IGV)[29] showed comparable signal to noise levels for H3K27ac CUT&Tag relative to ENCODE ChIP-seq, while H3K27me3 CUT&Tag exhibited consistently higher signal to background noise, in accordance with the improved FRiP scores (Fig. 3d). To correct for potential differences in breadth and read capture of ENCODE and CUT&Tag peaks, FRiP calculations were repeated for intersecting ENCODE ChIP and CUT&Tag peak regions, which were determined separately for each CUT&Tag sample (Supplementary Fig. 2c). This revealed that while H3K27ac FRiP scores were relatively similar for CUT&Tag and ENCODE, H3K27me3 produced approximately twice as many reads in overlapping intervals.

Since CUT&Tag and other Tn5 transposase-based methods may be susceptible to open chromatin bias resulting in preferential detection and over-representation of accessible regions of the genome[30], we assessed the proportion of reads falling into open chromatin ATAC-seq

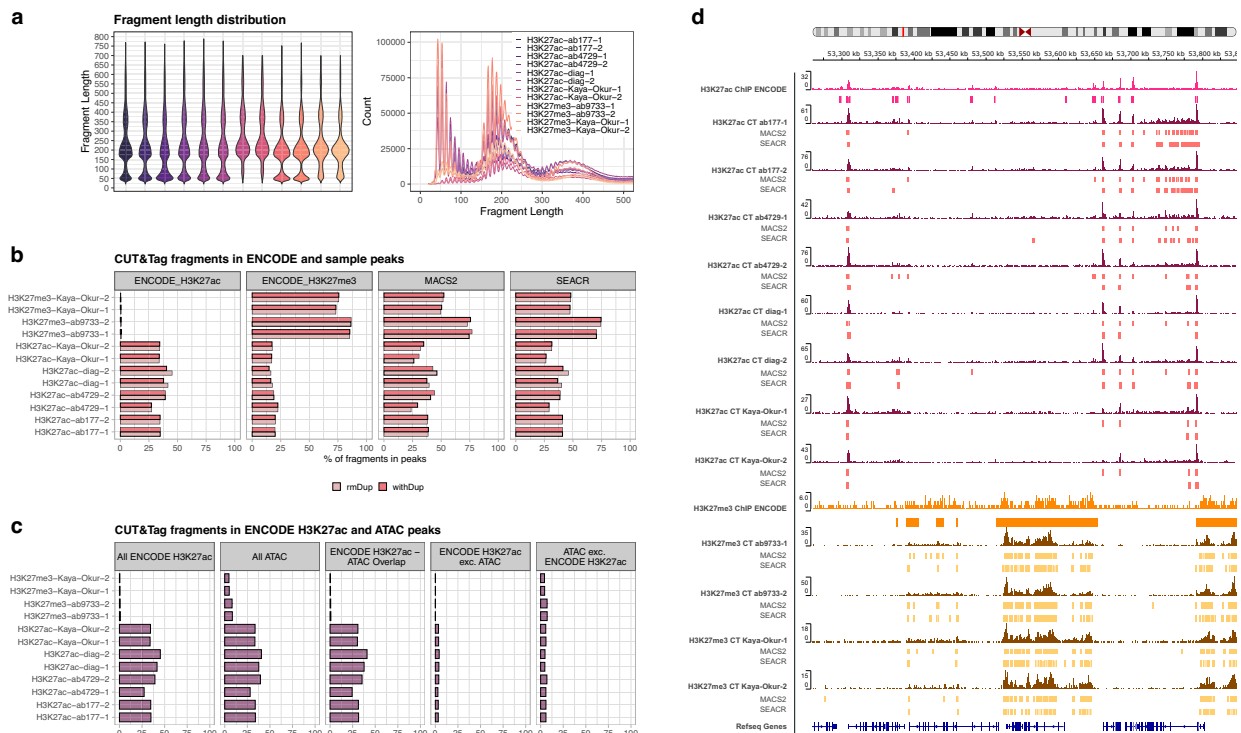

**Fig. 3 | Quality control metrics for CUT&Tag data. a** Violin and density plot of CUT&Tag fragment size distributions by sample. **b** Fractions of reads in peaks (FRiPs) defined by ENCODE for H3K27ac and H3K27me3 ChIP-seq, and peaks called for each sample with MACS2 and SEACR. **c**) Fractions of CUT&Tag sample fragments in ENCODE H3K27ac ChIP-seq and ATAC-seq peaks, as well as their overlapping and unique peaks by excluding (exc.) overlapping regions. **d** Integrative Genomics Viewer (IGV)[29] tracks showing ENCODE ChIP-seq and CUT&Tag (CT) read pileups and their corresponding MACS2 and SEACR peaks in a randomly selected genomic region (chr1:53,289,336-53,822,179). Pileup signals are shown in kilobases (kb).

peaks (Fig. 3c; Supplementary Fig. 2d). Nearly 70% of ENCODE H3K27ac peaks overlapped with ATAC-seq peaks. Therefore, ENCODE H3K27ac peaks were further subset to obtain those exclusive to the histone modification, exclusive to ATAC, and shared by both (Fig. 3c). This revealed that nearly all H3K27ac regions profiled by CUT&Tag fell into open chromatin regions shared with ATAC, but not into ATAC-only regions. For H3K27me3, around 8% of CUT&Tag reads fell into ATAC peaks (mean = 8.29, sd = 0.21), and this percentage was similar across comparison data (mean = 4.85, sd = 0.219). Removing short fragments (<100 bp) reduced this to 5.88% (sd = $4.82 \times 10^{-2}$). Short fragment exclusion had a similar effect on H3K27ac (mean = 35.60, sd = 4.61 to mean 28.31, sd=4.86), corresponding to a 29% to 20% reduction of CUT&Tag reads overlapping all ATAC peaks.

## CUT&Tag peak calling with SEACR and MACS2

We next assessed different peak callers and their settings to identify which would be most suitable for CUT&Tag. We evaluated the performance of both SEACR and MACS2, which were developed for CUT&RUN[28] and ChIP-seq[31], respectively. Parameter optimization was conducted based on precision and recall of ENCODE ChIP-seq peak capture by CUT&Tag (Fig. 4a; Supplementary Fig. 3a), with the aim of maximizing ENCODE capture while maintaining high precision (>75%). SEACR peaks were called using the stringent setting and thresholds of 0.01, 0.03, 0.05, or 0.1, as the relaxed setting was found to be too permissive, with precision scores consistently falling below the 75% threshold. MACS2 peaks were called using the narrow peak setting and p- and q-values (calculated using FDR correction) between $1 \times 10^{-5}$ and 0.1. These settings were also tested with local lambda deactivated to replicate the global background approximation employed by SEACR. Based on precision and recall analysis, optimum SEACR H3K27ac peaks were called using the stringent setting and a threshold of 0.01, and

narrow peaks with local lambda deactivated and a q-value of $1 \times 10^{-5}$ for MACS2. As a broader histone mark, H3K27me3 peaks were called with the same settings, but using the broad flag in MACS2 or an increased SEACR threshold of 0.1. With these parameters, SEACR peaks were called with slightly higher precision compared to MACS2, and antibody ab-177178 did not achieve the minimum 75% precision with MACS2, calling more peaks not identified in the ENCODE set (Fig. 4a; Supplementary Fig. 3b). CUT&Tag peaks identified with both peak callers were comparable in ENCODE recall at around 50%, capturing peaks of high signal intensity and missing some of lower intensities (Fig. 4a, b; Supplementary Fig. 3c). Peak calling was also attempted using the parameters mentioned above and published IgG control data. Although it increased ab-177178 precision, this did not improve ENCODE capture (Supplementary Fig. 3d) and was, therefore not used for further analyses.

SEACR defined a higher number of H3K27me3 peaks, but fewer H3K27ac peaks compared to MACS2. It also displayed robustness under very high duplication rates, while MACS2 called an excessive number of spurious peaks despite stringent parameters (Supplementary Fig. 3e). Although H3K27me3 peaks were of comparable width between peak callers, H3K27ac SEACR peaks were significantly wider, while the MACS2 peak width distribution more closely resembled that of ENCODE ChIP peaks (Supplementary Fig. 3f, g). The inclusion of duplicates did not affect peak widths. However, we note that increasing the stringency of the SEACR threshold from 0.1 to 0.01 results in a substantial increase in peak widths, thereby selecting peaks with more signal overall (Supplementary Fig. 3f, g). We observe that oftentimes multiple MACS2 peaks corresponded to a single SEACR peak (Figs. 3d, 4b; Supplementary Fig. 3b), ranging from an average of 1.35 to 1.68 (sd = 0.79-1.39) MACS2 peaks overlapping a SEACR peak per sample. While this may not pose an issue for broader histone marks, such as

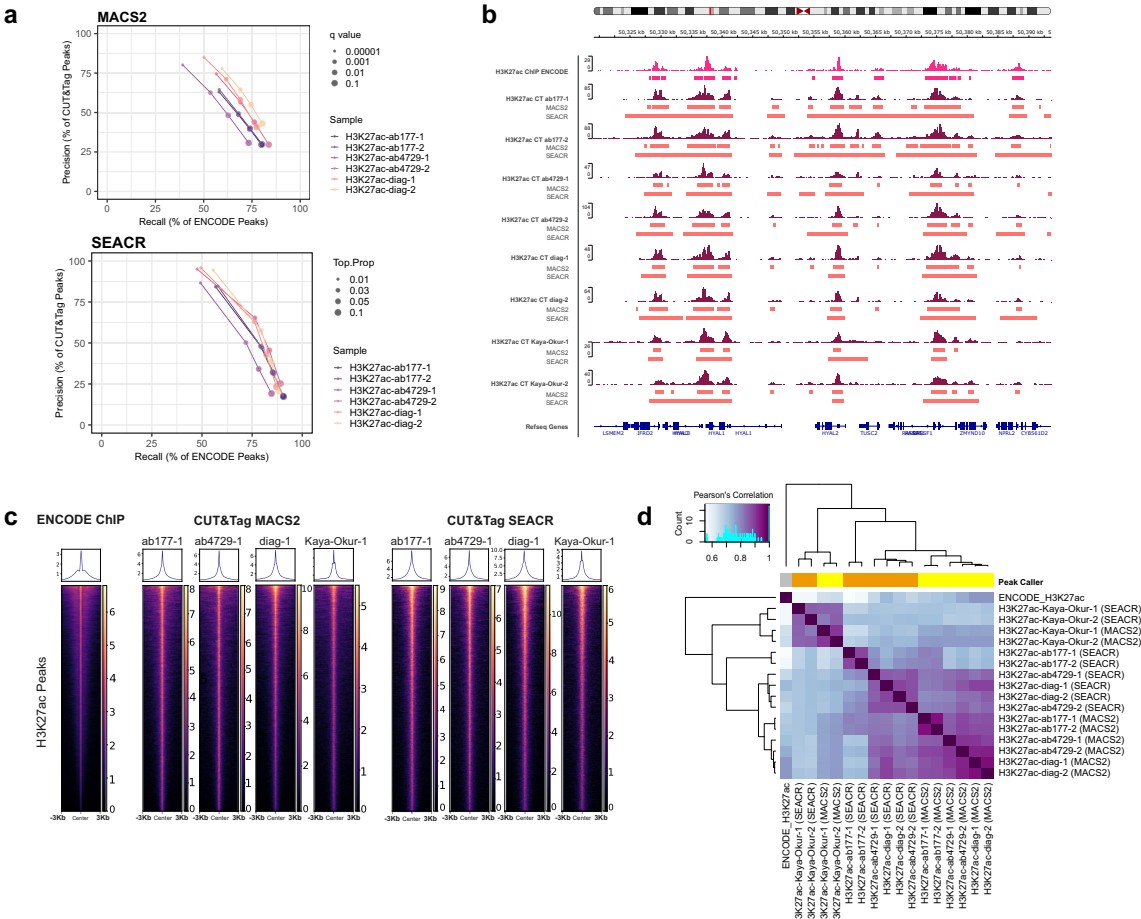

**Fig. 4 | H3K27ac peak calling with SEACR and MACS2. a** Precision and recall of MACS2 and SEACR peaks across peak calling parameters (MACS2: q-value; SEACR: proportion of top peaks). **b** Integrative Genomics Viewer (IGV) tracks for H3K27ac ENCODE ChIP-seq and CUT&Tag (CT) read pileups, alongside MACS2 and SEACR called peak ranges and Refseq genes. Pileup signals are shown in kilobases (kb). **c** Heatmap of H3K27ac read signal enrichments across a ± 3 kb window centered on peak summits from ENCODE H3K27ac ChIP-seq, and CUT&Tag MACS2 and SEACR, all subsampled to 2 million reads. Rows represent individual H3K27ac peaks and columns show signal values with lighter colors indicating higher signal intensity. **d** Clustered Pearson's correlation coefficients between H3K27ac sample peaks called with MACS2 (yellow) and SEACR (orange) created using DiffBind based on binary peak overlap.

H3K27me3, it may complicate the detection of subtle, local changes in histone modifications, and potentially merge nearby promoter and enhancer peaks.

Following subsampling to the same read depths, read profiles around peak summits confirmed that at the selected settings, peaks called by SEACR possessed broader read densities (Fig. 4c). CUT&Tag samples achieved greater peak read enrichment compared to ENCODE. However, this did not necessarily translate to higher FRiPs, which depend on abundance of peaks (Supplementary Fig. 3e). Genome-wide H3K27ac peak correlations, quantified with DiffBind[32], revealed that MACS2 called peaks with higher consistency than SEACR. MACS2 also possessed greater similarity to ENCODE peaks (which are also called using MACS2), particularly for the Diagenode antibody. Overall, both MACS2 and SEACR peaks were much more similar to ENCODE H3K27ac than H3K27me3 (Fig. 4d; Supplementary Fig. 4a).

### Benchmarking of CUT&Tag against ENCODE ChIP-seq
We proceeded to further benchmark CUT&Tag against ENCODE ChIP-seq profiles. First, in an attempt to minimize any bias potentially incurred by peak calling, samples were correlated on the basis of read counts in different genomic regions: ENCODE H3K27ac peak ranges (Fig. 5a; Supplementary Fig. 4b), the hg19 reference genome[33] partitioned into 500 bp bins (Supplementary Fig. 4c), and ENCODE H3K27me3 peak ranges (Supplementary Fig. 4d). While genome-wide

correlation revealed that similarity among CUT&Tag samples was markedly higher than that between CUT&Tag and ENCODE ChIP-seq or CUT&RUN, the CUT&Tag-ENCODE correlations were much enhanced when the analysis was restricted to ENCODE H3K27ac and H3K27me3 peak regions for the corresponding mark. This is likely because genome-wide comparison incorporates many regions that are devoid of true signal or contain noise, adding unwanted variability to the correlation analysis. Overall, there was high correspondence between read- and peak-level correlations (Figs. 4d, 5a; Supplementary Fig. 5a–c).

To determine the extent to which CUT&Tag recovers known ChIP-seq peaks, the GenomicRanges[34] package was used to calculate the proportion of ENCODE peaks overlapping with CUT&Tag (recall) and the proportion of sample peaks overlapping with ENCODE (precision; Fig. 5b; Supplementary Fig. 4e). Overall, ENCODE recall was comparable between MACS2 and SEACR, with an average of 54% across CUT&Tag experiments (sd. 4.99; Fig. 5b). The maximum ENCODE capture was 63% for Diagenode containing duplicates with the MACS2 peak caller. A slightly lower empirical ceiling was observed across the comparison of previously published CUT&Tag and CUT&RUN samples, with 41% for H3K27ac CUT&Tag with duplicates and MACS2 peak calling, and 53% for CUT&RUN with SEACR. Despite the much greater number of peaks called, H3K27me3 CUT&Tag reached an ENCODE coverage ceiling of approximately 58% (Fig. 5b). To test whether these

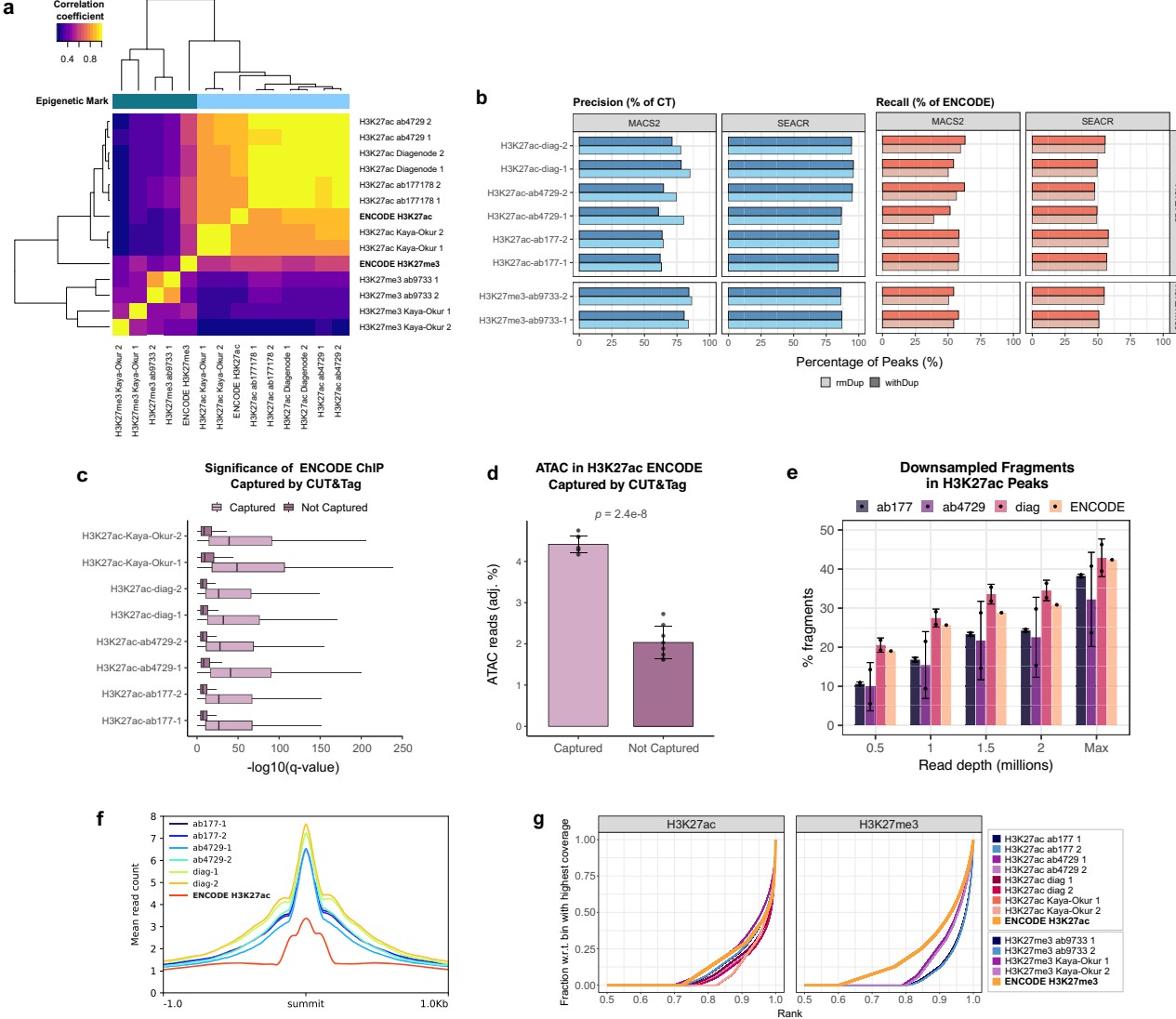

**Fig. 5 | Benchmarking of CUT&Tag against ENCODE ChIP-seq. a** Correlation of sample read counts across ENCODE H3K27ac peak ranges. **b** Percentages of CUT&Tag peaks falling into ENCODE H3K27ac (blue) and ENCODE H3K27ac peaks captured by CUT&Tag (orange), with and without duplicates. **c**) Comparison of -$\log_{10}$(q) values of ENCODE H3K27ac peaks captured and not captured by CUT&Tag peaks defined with MACS2; boxplot of median, first and third quartiles; whiskers correspond to 1.5 * the interquartile range (IQR). Welch two-sided t-test significance across all captured and missed peaks: $p < 2.2e-16$ (Ncaptured=196,698; Nmissed=212,710). **d** ENCODE ATAC-seq reads overlapping ENCODE H3K27ac ChIP-seq peaks captured and missed by CUT&Tag, as defined by MACS2. Reported as a proportion of total ATAC reads adjusted for genomic coverage of ENCODE peaks. P-value calculated using Welch two-sided t-test between captured and missed

peaks per sample ($n = 8$, including 6 in-house datasets of 3 antibodies with 2 replicates each, and 2 samples from Kaya Okur et al.). Error bars represent mean ± SD. **e** FRiPs of H3K27ac CUT&Tag sample peaks and ENCODE ChIP-seq peaks at different read depths, and at the original sequencing depth ('Max'). Data are represented as mean ± SD across antibody replicates ($n = 2$) and ENCODE ($n = 1$). **f** Average read counts around MACS2 peak summits of top H3K27ac CUT&Tag samples and ENCODE ChIP-seq peaks. **g** Fingerprint plots of H3K27ac and H3K27me3 CUT&Tag and ENCODE, all subsampled to 2 million reads. The bin with the highest coverage refers to the 1000 bp interval containing the most reads; graphs show the cumulative read counts within ranked 1000 bp bins, as a fraction of the read count in the highest-scoring bin.

metrics might be improved by further increasing library complexity, we generated aggregate, merged samples for either all internal H3K27ac CUT&Tag samples ($N = 6$), or merged by antibody ($N = 2$). Both approaches increased ENCODE recall, accompanied by a drop in precision. Precision was higher for peak sets from merged samples by antibody compared to all aggregated samples, although no MACS2 experiment reached 75% (Supplementary Fig. 4f). To determine whether these conclusions may be consistent across other cell types, we also assessed ENCODE precision and recall using published H3K27me3 CUT&Tag data from the HCT116 cell line[35] to find even poorer ENCODE capture at comparable levels of precision (Supplementary Fig. 4g).

To facilitate comparison between different samples, precision and recall were compounded into a single metric, the F1 score,

representing a weighted average of the two measures (Methods; Supplementary Fig. 4h). This approach excludes true negative peaks, which might distort the score since they occupy the vast majority of the genome. We confirmed Diagenode and Abcam-ab4729 as the better performing antibodies, while also finding that peak calling with SEACR resulted in slightly higher F1-scores on average. For antibody selection, ENCODE coverage was re-calculated with all samples subsampled to the same read depth. Here, antibodies were highly comparable, with the exception of one of the Diagenode samples showing improved metrics with SEACR peak calling (Supplementary Fig. 4h). Duplicates in CUT&Tag data may have biological relevance, potentially arising from tagmentation events that recur in the same place by chance. Thus, peaks were called with versus without duplicates.

Duplicate inclusion had no effect on SEACR precision or recall, and small increase in recall with decrease in precision when included with MACS2, although exacerbated when duplication rates are elevated (Fig. 5b, Supplementary Fig. 4e). Given the risks of detecting a large number of false-positive peaks at high duplication rates, we recommend exclusion of duplicates.

To further characterize the ENCODE H3K27ac peaks that were captured and missed in each CUT&Tag sample, the -log(q) significance values of the ENCODE peaks from original peak calling performed in ENCODE were compared. This showed that CUT&Tag captures the most significant peaks (Fig. 5c). We supplemented this by analyzing ATAC-seq read counts (Fig. 5d), as the H3K27ac marks should coincide with open chromatin regions. This showed that the ENCODE peaks captured by CUT&Tag samples contain more ATAC reads even when corrected for the total base count of the captured and missed ENCODE peak sets, supporting the notion that CUT&Tag detects more prominent H3K27ac peaks, or at least those that are more likely to also be detected by an orthogonal epigenomic method. In all cases, the differences between the q-values and ATAC-seq read counts in captured and missed ENCODE peaks were statistically significant ($p < 2 \times 10^{-16}$ across all q-value pairs and $p = 2.4 \times 10^{-8}$ for ATAC reads; two-tailed t-test).

Due to lower background signal, CUT&Tag should allow for higher data quality at read depths lower than those required for ChIP-seq, as previously shown for methyl histone marks[17]. To test whether H3K27ac CUT&Tag might have an advantage at lower read depths, FRiPs were calculated at 0.5, 1, 1.5 and 2 million unique reads (Fig. 5e). This analysis showed that H3K27ac CUT&Tag antibodies Abcam-177178 and Abcam-4729 produced fewer reads in peaks than ENCODE ChIP-seq at low read depth despite greater mean read counts around the peak summits (highest point of the peak detected by peak caller) (Fig. 5f). Only the Diagenode H3K27ac antibody produced higher FRiP scores than ENCODE when down sampled, and displayed the highest read pileups at the peak summit, in accordance with the highest peak calling precision scores (Fig. 5b). Cumulative sample read enrichments at equal read depths revealed that the read distributions of H3K27ac and H3K27me3 CUT&Tag samples showed more restricted read distributions than ENCODE H3K27ac and H3K27me3, respectively, though the effect was much more prominent for H3K27me3 (Fig. 5g). In agreement with the peak summit read enrichment analyses, this indicates that at equal read depths, H3K27ac CUT&Tag samples contain less off-target reads than ENCODE ChIP-seq.

## Functional analysis of CUT&Tag peaks

To investigate functional similarities of peaks identified by CUT&Tag compared to ENCODE ChIP-seq, we assessed the genomic distribution of CUT&Tag peaks in relation to genes and chromatin states. Using the ChIPseeker R package[36], peaks were mapped to their most proximal genes in terms of genomic distance to gene transcription start site (TSS). This revealed a strong skew towards promoter proximal regions for H3K27ac and a corresponding depletion in promoter regions for H3K27me3 (Fig. 6a). H3K27ac CUT&Tag peaks called by SEACR exhibited a stronger promoter preference than H3K27ac ENCODE ChIP-seq (Fig. 6a). H3K27ac CUT&Tag also showed similar enrichments for distal intergenic regions to ENCODE ChIP, which likely harbor a significant fraction of enhancers[37]. Next, we explored an alternative gene-independent breakdown of functional genomic elements by assigning peaks to ChromHMM-derived chromatin states[38] using the genomation R package[39]. This confirmed a predominance of promoters and enhancers amongst the regions mapped by H3K27ac CUT&Tag and ENCODE ChIP (Fig. 6b). In contrast, as expected, H3K27me3 overwhelmingly localized to heterochromatic and repressed chromatin regions. The two peak callers showed slight differences in regulatory element enrichment. While MACS2 corresponded better to ENCODE H3K27ac with enrichments at active promoters and strong

enhancers, SEACR peaks were additionally more enriched at weak enhancers, weak promoters, weakly transcribed regions, than their MACS2-called counterparts (Fig. 6b). This could be partly attributed to the fact that SEACR peaks are broader than MACS2 peaks, capturing more genomic sequence context and may extend to neighboring elements, as suggested by multiple state assignments for each SEACR peak.

Peaks specific to high duplicate-containing samples were functionally annotated to reveal that a significant portion of excess MACS2 peaks called upon inclusion of duplicates fall into heterochromatic regions, even among H3K27ac CUT&Tag samples (Supplementary Fig. 5a). This suggests that duplicates should not be retained when calling peaks with MACS2 in samples with high duplication rates, as it can lead to artifacts. On the other hand, the few extra peaks called by including duplicates in SEACR match the regulatory element distribution of the corresponding deduplicated peaks. Finally, CUT&Tag peaks that did not overlap with ENCODE spanned diverse element types, with tested antibodies showing an enrichment in areas of weak transcription, weak enhancers and heterochromatin, while published CUT&Tag and CUT&RUN data showed an enrichment for transcription elongation, weak transcription, weak enhancer and transcription transition categories (Supplementary Fig. 5b). For H3K27me3, these peaks were still almost exclusively located in heterochromatin regions. Read distributions around TSS obtained from NCBI RefSeq[40] were visualized with heatmaps, which showed enrichment around TSS for H3K27ac as expected (Supplementary Fig. 5c). When subsampled to the same read depth, CUT&Tag showed higher average read densities in these regions for H3K27ac compared to ENCODE ChIP and CUT&RUN. Reads from H3K27me3 samples generally did not co-localize with promoters, besides some residual CUT&Tag signal in our dataset. Enhancer enrichment was further tested by measuring capture of genome-wide STARR-seq peaks[41] (see Methods), at baseline and controlling for total genomic coverage (8 Mb). STARR-seq peaks were filtered to retain those that fall into K562 DNase-seq regions to yield putative cell-specific enhancers. ENCODE H3K27ac ChIP-seq recovered significantly more STARR-seq peaks than CUT&Tag, even when restricting to STARR-seq peaks overlapping with open chromatin regions that were profiled by DNase-seq experiments. However, when adjusting for genomic coverage overlap was highly comparable with MACS2 CUT&Tag (Supplementary Fig. 6a). This was accompanied by a shift towards higher q-value distributions of STARR-seq peaks captured versus missed by CUT&Tag and ENCODE (Supplementary Fig. 6b). To test whether SEACR promoter-overlapping peaks are more likely to also capture enhancers, we quantified the overlap with published STARR-seq data[41]. Calculating enhancer capture did not reveal a significant difference in promoter-enhancer co-occurrence among SEACR and MACS2 promoter-containing peaks (Supplementary Fig. 6a). A higher proportion of SEACR peaks overlapped promoter regions, although total promoter capture was comparable between peak callers, and in consideration with higher STARR-seq capture, it suggests SEACR peaks are less likely to be localized in enhancer regions (Supplementary Fig. 6c, d). Both peak callers displayed an average of one sample peak to capture promoter region (Supplementary Fig. 6e).

Finally, we performed gene ontology enrichment analysis visualizing the union of top fifteen pathways in each enriched category with the clusterProfiler R package[42]. Overall, CUT&Tag recovered all top ENCODE K562 H3K27ac ChIP-seq ontology terms, including cadherin binding, RNA-acting catalytic activity, and ubiquitin-like protein ligase binding (Fig. 6c). The correspondence of enriched terms indicates that although CUT&Tag may not recover all K562 ChIP-seq peaks, it performs sufficiently well to approximate the K562 regulatory landscape. We further conducted motif analysis with HOMER[43] using all peaks from each sample. Plotting the union of the top 15 enriched transcription factors per sample revealed that most CUT&Tag samples detected the ENCODE H3K27ac TFBMs. This includes those for the TFs

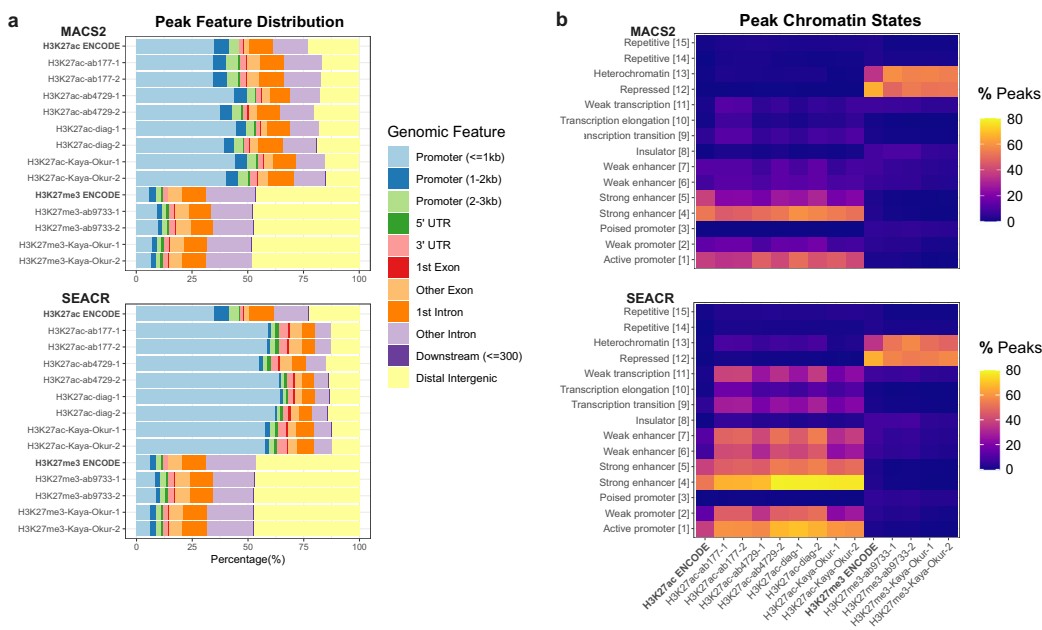

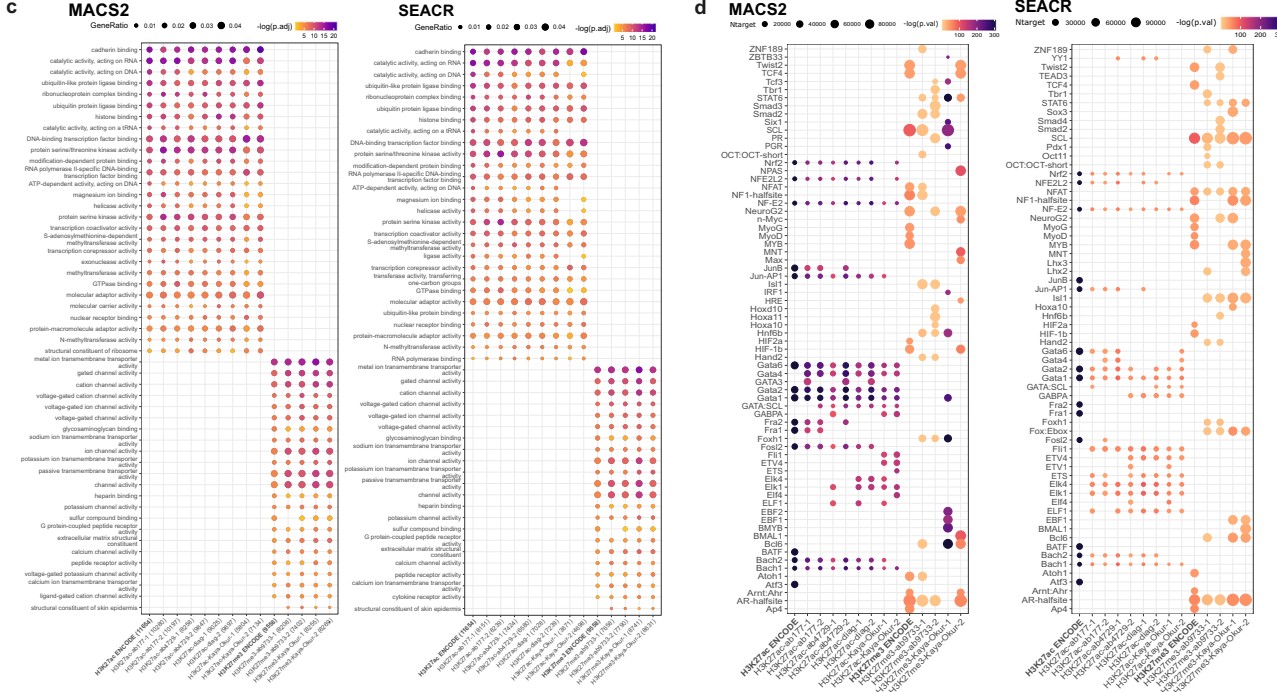

**Fig. 6 | Functional analysis of CUT&Tag and ENCODE ChIP-seq peaks.**
**a** ChIPseeker assignment of peaks to regulatory elements. **b** ChromHMM assignment of peaks to chromatin states, showing the relative percentages of total peaks falling into each category (note that peaks can fall into multiple categories simultaneously). **c** ClusterProfiler gene ontology enrichment analysis of genes assigned to sample peaks. The plot displays negative log-transformed p-values corrected for multiple testing correction with Benjamini-Hochberg at a p-value cutoff of 0.05 (-log10(p.adj)) for top enriched GO terms in the over-representation analysis using

a hypergeometric test. Dot size represents the GeneRatio, which reflects the proportion of total differentially expressed genes falling into a particular GO term. **d** HOMER top significantly enriched motifs across all samples. Dots colored by negative log-transformed p-values of enriched motifs from the hypergeometric test (-log(p.val)). Dot size corresponds to the number of times a motif appears in the target sequences (Ntarget). Motif enrichment was tested within 1000 bp windows of the peak center.

BACH1, BACH2, FOSL2, GATA1, GATA2, GATA6, JUN-AP1, JUNB, NRF2, NF-E2, NFE2L2, and NRF2 (Fig. 6d), which relate to cell growth[44] and hematological cell fate[45,46]. CUT&Tag also detected TFBMs not captured by ENCODE H3K27ac, such as those for ELK1, ELK4, GATA3, and GATA4, which are TFs involved in hematopoiesis[47]. This is consistent with the K562 lineage (lymphoblast) and was observed when calling peaks with both SEACR and MACS2 (Fig. 6d). H3K27me3 samples

showed more variable and modest TFBM enrichment, which is expected given that the vast majority of TFs bind in open chromatin regions.

## Discussion

Here, we optimized the execution and analysis of CUT&Tag for H3K27 histone marks, benchmarking its performance against matched

ENCODE ChIP-seq reference datasets. We studied H3K27ac in depth due to its functional co-localization with active promoters and enhancers, relevance for mapping risk variants in complex human disease, and the lack of previous literature optimizing CUT&Tag for acetylation marks. We systematically assessed experimental optimizations including antibody selection, antibody concentration, DNA extraction method, use of enzymatic inhibitors of deacetylases and PCR cycles. Due to the lack of consensus regarding specific analysis parameters, we assessed the performance of peak callers SEACR and MACS2 with different peak calling parameters and inclusion versus exclusion of duplicates.

Overall, H3K27ac CUT&Tag successfully recovers many features of ENCODE ChIP-seq and captures the most significant ENCODE peaks. However, across all individual samples, CUT&Tag only recovers around half of ENCODE peaks. Additionally, CUT&Tag appears to generate distinct peak profiles that favor H3K27ac domains coinciding with open chromatin regions and does not capture less significant ENCODE peaks, which are less enriched in open chromatin. It is uncertain whether this is a result of a failure to capture finer but nevertheless relevant ChIP-seq peaks or an indicator that ENCODE ChIP-seq may detect less relevant H3K27ac domains that have lower incidence of open chromatin, an important feature of active regulatory elements. Thus, although ENCODE ChIP-seq is often used as a standard reference dataset, it is unclear exactly how well ENCODE data reflects the ground truth. ChIP-seq peaks missed by CUT&Tag could potentially represent noise or false signals detected by ChIP-seq due to chromatin shearing and sonication, as well as fixation and cross-linking resulting in heterochromatin bias[13,14]. Investigation into new methods of chromatin profiling would significantly benefit from the inclusion of orthogonal approaches to mapping chromatin modifications and regulatory elements. We used K562 STARR-seq data[41] to estimate the capture of enhancer regions by both CUT&Tag and ENCODE and found that ENCODE recovers a greater number of putative enhancers with higher efficiency. However, since STARR-seq identifies regulatory elements independently of chromatin context, such analysis is ideally restricted to cell-specific active regulatory elements (by using ENCODE DNase-seq data). Benchmarking was also attempted with massively parallel reporter assay (MPRA) data[48], but the most significant regulatory assay quantitative trait loci (raQTLs) did not appear to be enriched in enhancer and promoter elements, and their coordinates could not be used as proxies for their genomic locations. Going forward, it would be important to determine whether H3K27ac ChIP-seq peaks that are not captured can be functionally validated.

The performance of CUT&Tag likely also varies depending on histone mark. It should be noted that the H3K27me3 antibody that was used in this study is monoclonal, whereas the H3K27ac antibodies used are polyclonal except for monoclonal Abcam-ab177178, which did not perform as well as polyclonal Abcam-4729. Enhanced performance of Abcam-4729 in H3K27ac CUT&Tag could relate partially to the fact that this is the antibody used by ENCODE ChIP-seq, leading to favorable results in ENCODE overlap and comparisons. Analysis of both current and comparison data showed that CUT&Tag achieves just over 50% ENCODE coverage for the H3K27ac mark, which may be improved to over 70% when using aggregate high-complexity samples. This suggests that joint peak calling across multiple merged samples may improve library complexity and signal capture, although at the expense of precision. CUT&Tag appears to perform better for methyl marks than for H3K27ac, and in the literature, the superiority of CUT&Tag over ChIP-seq was demonstrated on methyl marks[17]. However, it should be noted that our assessment of ENCODE capture by HCT116 CUT&Tag suggests that this might not always be the case. We found that peak calling parameters can greatly influence quality measures, such as FRiPs, and should be adjusted depending on histone modification. Nevertheless, it is uncertain why H3K27ac did not yield itself as well to CUT&Tag in this experimental context. A possibility is

that acetylation marks may be more dynamic (e.g. in response to environmental triggers) while methylation marks tend to be more stable[49]. It would be interesting to determine whether similar issues are encountered when profiling other acetyl marks, which has not yet been systematically addressed.

High duplication rates can result from overamplification during library preparation or over-sequencing. In either case, duplicates can be removed without compromising data quality, assuming there is an appreciable number of non-duplicate reads remaining. An advantage of high sequencing depths is sample saturation, meaning that the majority of unique fragments present in each sample was recovered. However, one intended advantage of CUT&Tag relative to methods such as ChIP-seq is the ability to recover comparable or superior levels of information at lower sequencing depths. Fractions of reads in ENCODE H3K27ac peaks were approximately equal with and without duplicates, suggesting that they are evenly distributed. Consequently, duplicates made little to no difference when calling peaks with SEACR, since genuine H3K27ac reads contributed to peaks that would in any case be called without duplicates, and reads outside genuine peaks did not meet the peak calling threshold. However, duplicates can result in the detection of a significant number of spurious peaks with MACS2 in samples with high duplication rates, many of which fell into heterochromatin regions, which should not be marked by H3K27ac[9]. Therefore, there is a marginal gain in recall (ENCODE capture) with the retention of duplicates at the expense of precision (detection of spurious peaks). Using fewer PCR cycles during library preparation appears to modestly reduce duplication rates without significantly influencing ENCODE recall.

Peak calling settings can have a significant effect on the outcomes of chromatin profiling experiments. In this study, multiple peak calling parameters were tested and selected based on precision and recall against matched ENCODE ChIP-seq profiles. MACS2 and SEACR performed similarly despite the marked differences in peak definition between the two peak callers. Which approach is most suitable is debatable because there is no strict definition as to what qualifies as a 'peak', but one concern is that peak calling with SEACR might make it difficult to detect subtle changes in histone marks due to its tendency to call wider peaks and combine multiple potentially distinct H3K27ac domains into single peaks. We did not find a significant difference in promoter-enhancer merging by the two peak callers, although SEACR appears to attain higher capture of weak enhancers and fewer sample peaks called outside promoter regions. With regards to differences in precision, MACS2 was optimized specifically for ChIP-seq. ChIP-seq samples are typically sequenced to much higher read depths and tend to possess higher levels of background, which is why MACS2 is designed to identify signals in data with high levels of noise[31,50]. In principle, CUT&Tag and CUT&RUN have reduced background as the only DNA fragments that are released are those bound by the protein of interest[17,51]. CUT&RUN is likely to be superior for mapping TF binding, as CUT&Tag employs elevated salt conditions for pA-Tn5 binding to prevent tagmentation of accessible DNA. This, in turn, can strip away transcription factor-DNA interactions, while histone modifications stay intact[21]. For a peak caller like MACS2, any off-target reads in samples with low background might be perceived as legitimate peaks, and this may explain why the inclusion of duplicates gave rise to spurious peaks. Notably, our analyses did not confirm a higher signal-to-noise ratio for H3K27ac CUT&Tag compared to ENCODE ChIP-seq profiles. Rather, H3K27ac CUT&Tag displayed equal or higher noise levels, in contrast to what was seen for H3K27me3 and other methyl marks in our analyses and previously reported by others. When restricting the interrogated regions to ChIP-seq and CUT&Tag-overlapping ranges to address the confounding effects of differential peak calling, H3K27ac CUT&Tag produced similar FRiP scores to ChIP-seq, whereas H3K27me3 CUT&Tag showed significantly improved performance. Whether this represents a general challenge for CUT&Tag of

histone acetylation marks remains to be explored. In the future, more specific peak calling methods designed for CUT&Tag data, such as the method GoPeaks[52], are likely to improve the performance of CUT&Tag profiling and should be included in future comparisons for benchmarking analysis.

The lack of established metrics to standardize performance makes it challenging to compare peak callers. Precision cutoffs are arbitrary, and there is an opportunity to significantly increase recall at the expense of precision, even within a predetermined boundary. Going forward, it may be worthwhile characterizing the CUT&Tag peaks that could be obtained without strict limits on precision to determine whether they could be legitimate peaks that are not captured by H3K27ac ChIP-seq. For example, it has been suggested that the relatively low correlation between CUT&RUN and ChIP-seq may be due to CUT&RUN's superior ability to map repetitive, difficult regions that are typically not covered by ChIP-seq[53]. We found some indication of this as CUT&Tag and CUT&RUN samples processed with SEACR were far more enriched in weak enhancers and weakly transcribed regions than ENCODE ChIP-seq, but MACS2 ChromHMM profiles differed minimally from ENCODE and this analysis indicated that this effect is more likely a result of peak caller selection rather than being intrinsic to CUT&Tag. One limitation associated with the use of ChromHMM annotations is that chromatin states are inferred on the basis of broad ENCODE ChIP-seq peaks, which introduces some circularity into overlap analysis with ENCODE H3K27ac peaks. Thus, chromatin states that occur in CUT&Tag but not in ENCODE H3K27ac are those that should theoretically not contain the H3K27ac mark. However, these annotations draw upon combinations of histone marks[54] and still give some indication as to where a particular modification might or might not be expected to occur.

The improved sensitivity of CUT&Tag compared to ChIP-seq is due to the use of pA-Tn5 to streamline library preparation through direct insertion of PCR sequencing adapters via in situ tagmentation. However, its sensitivity is inherently limited by PCR, since pA-Tn5 inserts its adapters in random orientations such that approximately half of the targets do not have adapters in the correct orientation to amplify. In addition, PCR library preparation is highly sensitive to size variations of amplicons. When two adjacent transposition events occur too far apart, they will not amplify efficiently during PCR or sequencing cluster generation. However, when they are too close, they will bias library coverage in an exponential manner due to increased PCR amplification and clustering efficiency of shorter fragments. One recent approach that may help overcome some of these issues is linear amplification by Targeted Insertion of Promoters (TIP-seq)[18,35]. Linear amplification appears to generate greater fidelity and uniformity, as mistakes made during amplification do not themselves become templates that can exponentially propagate errors. This results in higher mapping of single-cell sequencing reads[55]. Comprehensive optimization and benchmarking of this technique will be important moving forward.

CUT&Tag has been reported and widely adopted as a more streamlined, cost-effective approach to chromatin profiling. Despite a definite correspondence with ENCODE ChIP-seq, CUT&Tag consistently reaches an ENCODE recall ceiling of approximately 60%. Furthermore, the performance of this method appears to vary by histone mark. Additional analysis will be required to better characterize the inconsistencies between CUT&Tag and ENCODE ChIP-seq. Following optimizations of experimental parameters, we established Abcam-ab4729 and Diagenode as the top-performing antibodies for H3K27ac and demonstrated that the use of an HDACi does not improve H3K27ac CUT&Tag performance. Duplicates can and should be discarded, particularly beyond a threshold at which they start to contribute more off-target than on-target information. The optimal choice of peak caller is dependent on multiple input and output parameters. However, overall, MACS2 without the retention of duplicates seems to result in better performance metrics, more restrained peak widths, and slightly higher consistency. We observed that fewer PCR cycles reduced duplication rates at the expense of ENCODE recovery and capture. We hope that our systematic optimizations of CUT&Tag will help to facilitate its more widespread adaptation in the field and expedite its application in understanding the epigenetic causes and consequences of complex diseases.

## Methods

### Biological materials
Human K562 cells were obtained from ATCC (Manassas, VA, Catalog #CCL-243) and cultured according to the supplier's protocol. Mycoplasma was tested to be negative for all cellular input reported using Mycoplasma Detection Kit (Jena Bioscience PP-401) following manufacturer's instructions. The following antibodies were used: Guinea Pig anti-Rabbit IgG (Heavy & Light Chain) Preabsorbed antibody (Antibodies-Online ABIN101961), H3K27me3 (Cell Signaling Technology, 9733, Lot 14), H3K27ac (Abcam ab177178, Lot GR3202987-5), H3K27ac (Active Motif 39133, Lot 16119013), H3K27ac (Abcam ab4729, Lot G3374555-1), H3K72ac (Diagenode C15410196, Lot A1723-0041D). The following histone deacetylase inhibitors were used: Sodium butyrate (Merck B5887-250MG; used at 5 mM in CUT&Tag solutions with HDACi treatment), Trichostatin A (Enzo Life Sciences BML-GR309-0001; used at 1 μM in CUT&Tag solutions with HDACi treatment). The following commercial loaded protein A-Tn5 transposase fusion protein (pA-Tn5) were used at recommended dilutions by the manufacturer: CUTANA™ pAG-Tn5 (Epicypher 15-1017; Lot 20142001-C1), or pA-Tn5 Transposase - loaded (Diagenode C01070001; Lot 1/b/b).

### CUT&Tag nuclei processing
Bench top CUT&Tag was performed as previously described (https://www.protocols.io/view/bench-top-cut-amp-tag-bcuhiwt6)[17]. Exponentially growing K562 cells were harvested, counted, and centrifuged for 3 min at 600 g at room temperature (RT). 500,000 cells per condition were washed twice in 1 mL Wash Buffer (20 mM HEPES-KOH pH 7.5, 150 mM NaCl, 0.5 mM Spermidine, 1x Protease inhibitor cocktail; Roche 11836170001). Nuclei were extracted by incubating cells for 10 minutes on ice in 200 μL/sample of cold Nuclei Extraction buffer (NE buffer: 20 mM HEPES-KOH pH 7.9, 10 mM KCl, 0.1% Triton X-100, 20% Glycerol, 0.5 mM Spermidine, 1x Protease Inhibitor cocktail). Following incubation in NE buffer, nuclei were centrifuged for 3 min at 600 g at RT, then resuspended in 100 μL cold NE buffer. Concanavalin A-coated magnetic beads (Bangs Laboratories, BP531) were prepared as previously described[51], and 11 μL of activated beads were added per sample into PCR strip tubes and incubated at RT for 10 min. Beads were placed on a magnetic rack, and unbound supernatant was discarded. Bead-bound nuclei were resuspended in 50 μL Dig-wash Buffer (20 mM HEPES pH 7.5, 150 mM NaCl, 0.5 mM Spermidine, 1× Protease inhibitor cocktail, 0.05% Digitonin) with 2 mM EDTA and 0.1% BSA. Primary antibody was added at 1:50, 1:100, or 1:200 concentration and subsequently incubated on a rotating platform overnight at 4 °C. Primary antibody solution was removed by placing the PCR tube on a magnetic rack, allowing the solution to fully clear, then removing the supernatant. Next, the appropriate secondary antibody, Guinea Pig anti-Rabbit IgG antibody for a rabbit primary antibody, was added at 1:100 in Dig-Wash buffer and incubated at RT with rotation for 30-60 min. Nuclei were washed twice in 200 μL Dig-Wash buffer using a magnetic rack to remove unbound antibodies in supernatant. Nuclei were resuspended in 50 μL Dig-med Buffer (20 mM HEPES pH 7.5, 300 mM NaCl, 0.5 mM Spermidine, 1× Protease inhibitor cocktail, 0.05% Digitonin), then 1:20 CUTANA™ pAG-Tn5 (Epicypher 15-1017) or 1:250 pA-Tn5 Transposase - loaded (Diagenode C01070001) was added, gently mixed and spun down. pA-Tn5 binding occurred at RT for 1 hour on a rotating platform. To remove unbound pA-Tn5, nuclei were washed twice in 200 μL Dig-med Buffer. Nuclei were then resuspended in 50 μL

Tagmentation buffer (10 mM MgCl$_2$ in Dig-med Buffer) and incubated at 37 °C for 1 hour to activate transposase enzymatic activity. Next, either column or sodium dodecyl sulfate (SDS) based DNA extraction was conducted.

## Column DNA extraction

To stop tagmentation and solubilize DNA fragments, the following were added to each 50 μL sample: 1.68 μL 0.5 M ethylenediaminetetraacetic acid (EDTA), 0.5 μL 10% SDS, 0.44 μL 10 mg/mL Proteinase K. The samples were briefly mixed and vortexed at full speed for ~2 seconds, then incubated at 55 °C for 1 hour to digest the DNA. After a quick spin, tubes were placed on a magnetic rack, and the solution was allowed to clear. Supernatant was carefully transferred to a new 1.5 mL microcentrifuge tube, then the sample processing protocol of ChIP DNA Clean & Concentrator (Zymo Research D5205) was executed, eluting with 21 μL Elution Buffer.

## SDS-based DNA extraction

Following tagmentation at 37 °C for 1 hour, PCR tubes were placed on a magnetic rack, and the solution was allowed to clear. Supernatant was removed carefully, then beads were resuspended thoroughly in 50 μL [tris(hydroxymethyl)methylamino]propanesulfonic acid (TAPS) Buffer (10 mM TAPS pH 8.5, 0.2 mM EDTA) at RT. Tubes were returned to a magnetic rack, and the supernatant was removed. 5 μL SDS Release Buffer (10 mM TAPS pH 8.5, 0.1% SDS) was added at RT to each sample, and tubes were vortexed at full speed for ~10 seconds. After a quick spin, ensuring no beads are stuck to the side of the tubes, samples were incubated at 58 °C for 1 hour. Next, 15 μL SDS Quench Buffer (0.67% Triton-X 100 in Molecular grade H$_2$O) was added at RT and vortexed at maximum speed to neutralize the SDS prior to PCR library amplification.

## CUT&Tag PCR-based library amplification

For library amplification in PCR tube format, 21 μL DNA was combined with 2 μL of universal i5 and uniquely barcoded i7 primer[56] where a different barcode was used for each sample that was intended to be pooled together. 25 μL NEBNext HiFi 2× PCR Master mix was added, and then the sample was gently mixed through and spun down. The sample was placed in a Thermocycler with heated lid following these conditions: 72 °C for 5 min (gap filling); 98 °C for 30 s; 11–15 cycles of 98 °C for 10 s and 63 °C for 30 s; final extension at 72 °C for 1 min; and hold at 4 °C. Following PCR, bead cleanup was conducted by the addition of 1.1x Ampure XP beads (Beckman Coulter). Library and beads were mixed thoroughly, then spun down and incubated at RT for 10–15 min. Beads were gently washed twice with freshly prepared 80% ethanol using a magnetic rack, then the library was eluted with 20–30 μL 10 mM Tris-HCl pH 8.0 at RT.

## Sequencing

Final library size distributions were assessed by Agilent 2100 Bioanalyzer and Agilent 4200 TapeStation for quality control before sequencing. Libraries were pooled to achieve equal representation of the desired final library size range (equimolar pooling based on Bioanalyzer/TapeSation signal in the 150 bp to 800 bp range). Paired-end Illumina sequencing using the HiSeq 4000 PE75 strategy was conducted on barcoded libraries at Imperial Biomedical Research Centre (BRC) Genomics Facility following manufacturer's protocols.

## qPCR

Quantitative real-time PCR (qPCR) was performed following manufacturer's instructions in triplicate technical and triplicate biological replicates (https://www.thermofisher.com/order/catalog/product/4309155#/4309155). Positive and negative control primers were designed based on ENCODE peaks ranked highest to lowest, respectively (Supplementary Table 1). Levels of H3K27ac CUT&Tag binding

signal was determined by qPCR amplification carried out with the QuantStudio™ 5 Real-Time PCR System (ThermoFisher A34322) using the Standard Curve experiment type and SYBR Green Master Mix (ThermoFisher 4309155). Each qPCR condition was conducted with triplicate repeats and the data was analyzed using the $2^\wedge$-$\Delta\Delta CT$ method where each CUT&Tag sample was normalized to qPCR levels of K562 genomic DNA (gDNA) run in parallel. qPCR results were calculated using the equation:

$$2^{-(CT\,sample - CT\,DNA)} \tag{1}$$

## Data processing

The full dataset, i.e. paired-end reads were used for the analysis. Sequencing data was processed according to the CUT&Tag Data Processing and Analysis Tutorial (https://yezhengstat.github.io/CUTTag_tutorial), with some alterations. Raw sequencing reads were trimmed using TrimGalore (version 0.6.6; https://github.com/FelixKrueger/TrimGalore) to remove adapters and low-quality reads. The trimmed fastq files were aligned to hg19 using bowtie (version 2.2.9[57]) with the following parameters: --local --very-sensitive --no-mixed --no-discordant --phred33 -I 10 -X 700. PCR duplicates were removed using Picard (version 2.6.0; http://broadinstitute.github.io/picard/), and bam and fragment bed files from original and deduplicated alignments were generated using samtools (version 1.3.1[58]) and bedtools (version 2.25.0[59]), selecting for fragment lengths under 1000 bp. Peaks were called using MACS2 (Model-based Analysis of ChIP-seq; version 2.2.9.1)[50] and SEACR (Sparse Enrichment Analysis for CUT&RUN; version 1.3)[28]. MACS2 peaks were called as follows: macs2 callpeak -t input_bam -n sample_name -f BAMPE -g hs -q 1e−5 --keep-dup all −nolambda --nomodel --outdir out_dir. SEACR peaks were called on the basis of fragment bedgraph files generated with bedtools genomecov. SEACR peaks were called as follows: SEACR_1.3.sh input_bedgraph 0.01 non-stringent out_name. In both cases other combinations of peak calling settings were also tested (see **Results**). To test peak calling performance with control data, published IgG CUT&Tag replicate libraries (SRR8754611 and SRR8754612; study accession PRJNA512492)[17] were downloaded in fastq format from the European Nucleotide Archive[60] (https://www.ebi.ac.uk/ena/browser/home), pooled together, and processed as described to produce bam and bedgraph files as inputs for MACS2 and SEACR; otherwise, peaks were called using previously selected parameters. Motifs were identified from complete peak sets using HOMER (Hypergeometric Optimization of Motif EnRichment; version 4.11.1[43]) as follows: findMotifsGenome.pl input_bed hg19 out_dir -size 1000. Down sampled bam files were generated by random sampling of original bam files as follows, where {x} represents the seed value and {y} the fraction of total read pairs to be sampled: samtools view -bs {x}.{y} input_bam > downsampled_bam.

To more closely approximate the replicated peaks strategy employed by ENCODE, pooled peaks were called with SEACR and MACS2 across thresholds after merging bam files across all CUT&Tag H3K27ac samples generated in this study (6 samples total) or antibody replicates (2 samples per antibody). This procedure was carried out by running the pooled_peaks function in the R package PeakyFinders (https://github.com/neurogenomics/PeakyFinders). EpiCompare[61] was then run on each set of pooled peak files separately to generate precision-recall curves.

## Sample comparisons

Published CUT&Tag[17] and CUT&RUN[22] samples were obtained as fastq files from the European Nucleotide Archive (study accessions PRJNA512492 and PRJNA522731, respectively) and processed as described above. Peak-level correlations were obtained with the DiffBind package (version 3.8.4[32]). Genome-wide sample correlations were carried out using bedtools multicov against hg19 split into 500 bp bins.

Read counts were then quantile-normalized and rounded to the nearest integer, and heatmaps plotted in R[62] based on sample-by-sample Pearson correlations of the processed counts. Fingerprint plots were generated from sample and ENCODE bam files using deepTools (version 3.5.4[63]) plotFingerprint, setting genome-wide bin sizes of 1000 bp. Heatmaps were plotted using deeptools computeMatrix and plotHeatmap to visualize read enrichment around hg19 transcription start sites (obtained from NCBI RefSeq) and peak summits. For these ends, ENCODE H3K27ac and H3K27me3 samples (ENCSR000AKP and ENCSR000EWB, respectively) were run through the ENCODE histone ChIP-seq pipeline (https://github.com/ENCODE-DCC/chip-seq-pipeline2), with replicates down sampled to 1 million reads per sample and pooled together. Likewise, to plot heatmaps, paired-end CUT&Tag sample bam files were down sampled to 2 million fragments (4 million reads) and only the first of the read mates mapped to yield a total of 2 million mapped reads. As a weighted average of precision and recall, F1-scores were calculated as follows, where $t_p$, $f_p$, and $f_n$ represent the numbers of true positive, false positive, and false negative CUT&Tag peaks, respectively:

$$F1 = \frac{t_p}{t_p + 1/2\left(f_p + f_n\right)} \tag{2}$$

### Downstream data analysis

Downstream analysis, including quality control, ENCODE benchmarking, and regulatory element annotation, was performed in R[62]. Peak overlaps were determined with the GenomicRanges package (version 1.50.2[34]). All peaks overlapping with hg19 blacklisted regions (ENCODE ID: ENCFF000KJP) were removed prior to downstream analysis. Peaks falling into mitochondrial or other non-standard chromosomes were also excluded using BRGenomics (version 1.10.0; https://mdeber.github.io). FRiP scores were calculated using the chromVAR package (version 1.20.1[64]). ATAC-seq libraries (ENCODE IDs: ENCLB918NXF and ENCLB758GEG) used for FRiP analysis were processed with the nf-core ATAC-seq pipeline[65]. To calculate FRiP scores in overlapping ENCODE and CUT&Tag peak regions, CUT&Tag peak ranges intersecting with ENCODE H3K27ac replicated narrow peaks (ENCFF044JNJ) and H3K27me3 replicated broad peaks (ENCFF000BXB) were obtained using the bedtools intersect tool; the same peak files were used for ENCODE capture calculations. This was performed for each sample-ENCODE pair. For the calculation, pooled reads from ENCODE H3K27ac ChIP-seq replicates ENCFF384ZZM and ENCFF070PWH, and H3K27me3 ChIP-seq replicates ENCFF000BXA and ENCFF000BXC, were used. To determine the precision and recall of ENCODE peak capture in the HCT116 cell line, H3K27me3 narrow peaks (ENCFF255ARD) and published H3K27me3 CUT&Tag data (study accession: PRJNA779107; sample accession: SRR16963158) were used[35]. Regulatory element annotation was performed using ChIPseeker (version 1.34.1[36]), after annotating peaks with genes using the TxDb.Hsapiens.UCSC.hg19.knownGene database (version 3.2.2[66]), tssRegion -3000, 3000, and flankDistance 5000. To identify promoter-overlapping peaks, this database was also used as input to the ChIPseeker getPromoters function to obtain genomic ranges 1.5Kb upstream and downstream of known promoters. ChromHMM annotations assigned with genomation (version 1.30.0[39]). To test for enhancer enrichment, STARR-seq peak files (ENCFF717VJK and ENCFF394DBM) were processed to keep only replicated peaks and lifted over to hg19 using the rtracklayer liftOver function[67]. To obtain putative cell-specific enhancers, STARR-seq peak ranges were filtered to keep those overlapping with K562 DNase-seq peaks (ENCODE ID: ENCFF722NSO). Functional enrichment analysis was carried out with clusterProfiler (version 4.6.2[68]), using the "enrichGO" function. Results from motif analysis were processed with marge (version 0.0.4.9999, https://github.com/robertamezquita/marge). For visualization with the Integrative Genomics Viewer (IGV; v2.9.4)[29], sample bedgraphs were converted to bigwig files using the UCSC Genome Browser bed-GraphToBigWig binary. Single ENCODE replicate bam files (H3K27ac: ENCFF384ZZM; H3K27me3: ENCFF000BXA) were converted to bedgraphs with bedtools genomecov[59], and similarly converted to bigwig format.

### Reporting summary

Further information on research design is available in the Nature Portfolio Reporting Summary linked to this article.

## Data availability

The CUT&Tag data generated in this study have been deposited in the Gene Expression Omnibus (GEO) database under accession code GSE286492. The bigwig, bedgraph, and peak data generated in this study can be found at: https://data.cyverse.org/dav-anon/iplant/home/paulinaurbana/H3K27_CUT%26Tag_Benchmark/. The UCSC bigwig tracks data generated in this study can be found at: https://genome.ucsc.edu/s/pu1918/CUT_and_Tag_benchmarking. Published IgG CUT&Tag libraries were accessed from SRR8754611 and SRR8754612. We downloaded published H3K27ac and H3K27me3 CUT&Tag sequencing datasets from BioProject accession code PRJNA512492, and CUT&RUN from PRJNA522731. Comparisons were made against ChIP-seq datasets accessed from ENCODE experiments ENCSR000AKP and ENCSR000EWB for H3K27ac and H3K27me3, respectively. Blacklist regions for filtering peaks were used from ENCODE file ENCFF000KJP. ATAC-seq libraries were accessed from ENCODE experiments ENCLB918NXF and ENCLB758GEG. ENCODE H3K27ac and H3K27me3 ChIP-seq peak sets accessed from ENCFF044JNJ and ENCFF000BXB. HCT116 cell line H3K27me3 ChIP-seq narrow peaks were downloaded from ENCODE ENCFF255ARD, and CUT&Tag data for sample SRR16963158 was accessed from BioProject PRJNA779107. Enhancer STARR-seq peaks were accessed from ENCFF717VJK and ENCFF394DBM, which were overlapped with K562 DNase-seq peaks from ENCFF722NSO. Source data are provided with this paper.

## Code availability

Code used in this study is available in the dedicated GitHub repository: https://github.com/Marzi-lab/CUTnTag-benchmarkinganalysis and on Zenodo[69]. Generalized code for performing comparisons between genome-wide histone modification profiles has been made available in the EpiCompare R package[61] via GitHub at https://github.com/neurogenomics/EpiCompare.

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

## Acknowledgements

We thank the UK DRI Neurogenomics Laboratory, in particular Alan E Murphy, for feedback, troubleshooting, and guidance in experimental design and analysis. We thank Katiuska Pulgar Prieto for help with K562 mycoplasma testing. SJM and AN were supported by the Edmond and Lily Safra Early Career Fellowship Program. SJM, NGS, and AN are supported by the UK Dementia Research Institute [award numbers UKDRI-6205, UKDRI-5008, and UKDRI-5016] through UK DRI Ltd, principally funded by the Medical Research Council. SJM received funding from the Alzheimer's Association (grant number ADSF-21-829660-C) and the MRC (grant number MR/W004984/1).

## Author contributions

All authors conceived the study. DH and JNI performed all laboratory experiments. LA and PU performed computational analyses with the guidance and support of BMS. SJM and NGS acquired funding and supervised the study. AN guided study directions and interpretations. All authors wrote, read, and approved the manuscript.

## Competing interests

The authors declare no competing interests.
