## [Transparent Peer Review file · Nature Communications]

CUT&Tag recovers up to half of ENCODE ChIP-seq histone acetylation peaks

Corresponding Author: Dr Sarah Marzi

Version 0:

Reviewer comments:

Reviewer #1

(Remarks to the Author)

Hu and Abbasova et al present a thorough comparison of CUT&Tag to ChIP-seq from the ENCODE consortium reaching several major conclusions. 1) CUT&Tag can identify up to 50% of peaks present in ENCODE, 2) That SECAR has superior operating characteristics to MACS2, 3) that duplicates should be omitted.

This sort of comprehensive optimization and validation work is an important resource for the community. Indeed, there is not a standardized workflow for CUT&Tag or quality metrics (as discussed below), which makes the comparison of datasets generated in different labs challenging. I appreciate the authors efforts to present the optimization work they have done and to provide a standardized workflow.

However, my major concern is that the complexity of the H3K27Ac CUT&Tag libraries appears to be quite low with duplication rates frequently being in the high 90% range (Supplementary Table 1, Figure 6a). In the DNA isolation experiments, duplication rates appear lower (50-70% range), but it does not appear that these experiments were the ones stringently evaluated in the prior figures. In our experience, high quality CUT&Tag libraries for H3K27Ac (using ab4729) will have duplication rates in the range of 50-70%, but will often times be much lower and rates of <20% for H3K27me3 are quite common. As a result, the conclusions drawn in this study may not be comparable to what others will find in more complex datasets. The consequences of retaining duplicates in this dataset as compared with a dataset with a 40% duplication rate will be very different.

An explanation for the low complexity of these libraries isn't completely clear. We have had good luck with H3K27Ac profiling in leukemia cell lines using ab4729 at 1:50 and the CUT&Tag@home protocol without digitonin, and routinely get duplication rates of 50%. We have found that using LowBind PCR tubes, minimizing pipette mixing or inversion of the tubes leads to maximal bead/cell retention and higher complexity libraries. Some have reported on the protocols.io comments section for the CUT&Tag protocol, that certain H3K27Ac antibodies work better in some cell lines than in others. Indeed, the Kaya-Okur CUT&Tag for H3K27Ac in K562 cells does not have exceedingly high signal to noise (based on our groups internal analysis of this data), so this may be a cell line specific issue. We have not evaluated K562s but have had reasonable luck with most leukemia cell lines and primary hematopoietic cells we have tried.

Major points:

1. Comparison of CUT&Tag with Encode

- a. As the authors performed CUT&Tag for only two histone marks and ENCODE covered many histone marks in addition to transcription factors, it would probably be better to have a more specific title (i.e. ENCODE H3K27Ac and H3K27me3 peaks) and conclusions as this work did not cover all of ENCODE.
- b. The low complexity of the libraries precludes making definitive statements about whether CUT&Tag can satisfactorily capture ENCODE peaks.
- c. With high complexity libraries we frequently sequence deeper (10M reads) and retain duplicates which results in higher peak numbers and improves power in differential binding experiments. It would be interesting to see if higher complexity libraries sequenced deeper would capture a greater percentage of ENCODE peaks.
- d. Perhaps the authors could repeat these studies on another cell line used in ENCODE, generate higher complexity data and generate more generalizable conclusions?

2. Comparison of SEACR to MACS2

- a. Another issue that is not comprehensively addressed is that the peaks produced by SEACR are very wide and promoter peaks frequently are merged with nearby enhancers in our experience. This can have significant downstream ramifications, depending on goals of secondary analysis. It is often preferable to have narrower peaks when performing differential peak calling and motif enrichment studies. In these instances, a peak caller that calls narrower peaks may be more appropriate.
- b. MACS2 may perform better on higher complexity datasets.
- c. Other peak callers have been developed (as cited by the authors) which have not been benchmarked here.

3. Retention of duplicates

- a. As described above, this will be highly dependent on the degree of duplication and more complex libraries will be less impacted by duplication. In this setting, retention of duplicates may be appropriate.

Minor Points

- The discussion about the brain in the introduction seems out of place considering all the data presented are in a leukemia cell line
- The fonts in the figures are frequently very small and difficult to read
- More closeup images of tracks and called peaks would enable readers to better assess data quality
- Listing peak number in addition to percentage overlap with Encode would be valuable
- Please include the ENCODE ID in the text and methods.
- Please include the source of the ATAC-Seq dataset. Is it from ENCODE too?
- For SEACR peak calling, did you use any control samples like IgG?
- For MACS2 peak calling, did you use any control samples like IgG?
- Figure 7a and 7b are blurry
- What is the dashed line in Figure 4G?
- Line 435: Please include number of peaks fed into HOMER motif analysis for each method to highlight any differences/similarities since that can really affect the magnitudes of HOMER p-values.

Theodore P. Braun MD, PhD
Assistant Professor
Division of Hematology & Medical Oncology
Division of Oncologic Sciences
Knight Cancer Institute
Oregon Health & Science University

Reviewer #2

(Remarks to the Author)

The authors optimize CUT&Tag and benchmark it against ChIP-seq. In light of the rapidly growing popularity of CUT&Tag for low cell numbers and single-cells, this study is very timely. Also, there is a lot of useful information that users will find helpful, such as what antibody concentrations to use and showing that consistent results are obtained for an acetyl mark in the presence or absence of HDAC inhibitors. Although I have no criticisms of the experimental aspects of the work or the functional annotations, I found flaws in the analysis that undermine the major conclusion that is the title of the paper. The comparisons that the authors do are based on peak calling, but their analysis did not sufficiently compensate for differences between CUT&Tag and ChIP-seq data. There is no ground-truth set of peaks given the vagaries of peak finders - they will all find peaks in high-signal regions and different sets of noise in low-signal regions, and showing that different peak callers or parameters gives similar results doesn't solve the problem. Further analyses are needed to make this part of the manuscript suitable for publication in Nature Communications.

Major issues

1) The observation that CUT&Tag shows a higher concentration of reads over read summits than ENCODE ChIP-seq can explain why it has lower FRiP values than ChIP-seq, because ChIP-seq peaks are wider and so will capture more reads all else being equal. This is evident from Figure 4f, where the narrower CUT&Tag peaks indicate better resolution than ChIP-seq. In FRiP analysis, the tighter concentration of CUT&Tag reads over the summit is a disadvantage, and it casts doubt on the value of all the FRiP comparisons between CUT&Tag and ChIP-seq in the paper. One way to address this issue is to compare their sets of peaks in a peer-to-peer manner, for instance using the bedtools intersect tool.

2) The most obvious interpretation of the observation that CUT&Tag captures the most significant peaks but misses the less significant peaks (Fig. 4c) is that the less significant peaks are false positives as a result of noise, and this is understandable in part because of the broader ChIP-seq peaks using the same peak-caller pointed out above. In other words, the broader peaks of ChIP-seq may be broad because the overall background is higher, as previous work has shown (but in contrast to statements in the paper), and this makes it difficult for a peak-caller to distinguish true from false positives. This possibility is suggested by the graph in Figure 4f, where the CUT&Tag curves are still descending at -0.5 and +0.5 kb, whereas the ENCODE ChIP-seq curve had completely leveled off by about ± 0.2 kb. The fact that the CUT&Tag curves intersect the ChIP-seq curve at ± 0.5 kb and are still descending is clear evidence that the CUT&Tag background levels are lower genome-wide and provides support for the above explanation. At a minimum the authors need to extend the horizontal axis to the point that the CUT&Tag data levels off. This interpretation is also consistent with the finding of higher read density for CUT&Tag over TSSs (Figure S5e). More to the point, unless the authors can rule out this explanation, the title of their paper "CUT&Tag recovers up to half of ENCODE ChIP-seq peaks" is simply explained by the higher background of ChIP-seq. See my suggestion above to help address this issue.

3) The authors state in the Methods section that they used only one end of the paired-end read data for CUT&Tag, presumably because the ChIP-seq data was only single-end. The authors must suppose that using only half of the CUT&Tag data is fair, but it is not, as the point of doing paired-end reads is to get the precise position of the fragment, whereas the ENCODE Project purposely did only single-end reads because it was cheaper and faster than paired-end sequencing and there was little justification given the low resolution of ChIP-seq. Here the authors did PE75 sequencing and used this to plot the length distribution in Fig. 2b, and so to use the data properly they should align the fragments and use that for peak calling and FRiP analysis to see how well CUT&Tag can do when performed as intended. Otherwise, they are degrading the CUT&Tag data in a way that users will not. This not only biases the result in favor of ChIP-seq, but is totally unrealistic, as researchers wouldn't think of throwing away half of the data.

4) Since the authors have paired-end data they should use it like any researcher would do. SEACR requires a track (bedgraph file) as input, but MACS2 prefers a bed file of mapped fragments (BEDPE option with PE data). It can perform model-building when given mapped fragments instead of a track. In all cases, fragments mapped to chrM should be removed before peak-finding as they will skew a model-based peak finder.

5) The text referring to Figure 2c is confusing. The text states that "all H3K27ac CUT&Tag FRiP scores were markedly lower than the ENCODE H3K27ac ChIP-seq score of 42%" but this contradicts the previous sentence, and in Figure 2c all of the histogram bars are below 42%. If anything the opposite seems to be the case: For example using Abcam 4729 CUT&Tag peaks are ~35% whereas ENCODE peaks are ~20%. The wording in the text is perhaps ambiguous, in that one might conclude that the comparison is done by calling ENCODE peaks and calculating the fraction of ENCODE and CUT&Tag reads in the ENCODE peaks, which would be trivial, like saying a house is more like itself than it is like the house across the street. Regardless, there needs to be some clarification, since FRiP comparison is how data quality is judged, and the issue raised by the title of this article suggests that there is a data quality issue that the study tries to resolve. FRiPs should be used with extreme caution when comparing different peak finders because just producing more peaks or wider peaks regardless of quality will result in a higher FRiP value.

Minor issues

I found misleading the use of the term "gold standard" in the Abstract and elsewhere to be somewhat misleading. This term is generally understood as closest to ground truth: "the best, most reliable, or most prestigious thing of its type." But setting up ENCODE as the gold standard is prejudicial, because only after the results of the comparison are evaluated can it be decided which if either should be considered the gold standard, and even then it's tricky to make a call. Calling ENCODE ChIP-seq "a de facto gold standard" or "the current gold standard" is better, but just "current standard" would be more accurate.

The authors need to state whether or not the antibodies used were the same between ENCODE and CUT&Tag. Checking GenBank shows that this was the case for both H3K27ac (Abcam ab4729) and H3K27me3 (CST 9733), which is good. However, whereas the H3K27me3 antibody is monoclonal, the Abcam H3K27me3 antibody is polyclonal, and this complicates a direct comparison. Since multiple antibodies were used for H3K27ac and ab2729 was one of the best, this is unlikely to be a major issue, but this caveat should be mentioned.

Reviewer #3

(Remarks to the Author)

This article showed the performance of CUT&Tag and comparison with ENCODE ChIP-seq. The epigenomic profiles like histone modifications can be interpreted by dissecting DNA-protein interactions using next generation sequencing. The authors generated their own 30 datasets by the CUT&Tag method with multiple commercially popular antibodies against H3K27ac and H3K27me3. They performed a massive computational analysis to benchmark CUT&Tag data against ChIP-seq data from ENCODE. The result is mostly consistent with the previous notion; CUT&Tag recovers no more than half of ChIP-seq peaks. They tested library preparation method (PCR cycles and DNA purification method), compared two peak callers with different parameters, and checked precision and recall rate to optimized the best condition.

Overall, the idea and flow of this article are quite informative and invoke interest for ones seeking the epigenomic features. However, the original CUT&Tag technique was published in 2019 and afterwards multiple papers were used this method to show different applications. The analysis pipelines like SEACR and MACS2 were developed by other groups. The only progress is a systematic comparative analysis of two analysis pipelines using CUT&Tag and ChIP-seq datasets. Therefore, I think the important advances of significance to epigenetic specialist seem to be relatively weak. The authors did not address quite well the question why CUT&Tag could not recover the true ChIP-seq peaks. Instead they emphasized only the benchmarking result and optimization.

There are many points to be considered.

1. The nature of H3K27ac and H3K27me3 might not be properly appreciated; H3K27ac shows generally narrow point peak profiles and H3K27me3 broad peak patterns. Direct comparison of these two histone modification profiles is expected to be incompatible.. As you know, the histone modification of H3K4me3 is usually enriched at the promoters and show the similar narrow peak shapes like that of H3K27ac. Rather than H3K27me3, H3K4me3 would give a better result to support your benchmarking outcome of H3K27ac CUT&Tag

2. The enzymatic activity and efficiency of Tn5 affect the data quality for CUT&Tag which is principally different from ChIP-seq where the antibodies bound to target proteins are pulled down. That's why the discrepancy between CUT&Tag and ChIP-seq appears. I think it would be better to interpret your data as a complementary cooperation between two methods to interpret the biologically natural epigenetic modifications.

3. The figures should be reorganized. For example, Fig 1 can be moved to supplementary section because the experimental methodology and analysis pipelines are not original and just help readers for easy understanding your analysis. Fig 3a and Supp. Fig2c are overlapped. And many parts in main Figures are partly overlapped with the supplementary figures, which are redundant and need to be simplified.

4. In Fig.4b, there is no data for with and without duplicates.

5. The section, "Experimental optimisation of CUT&Tag" is a minor point in this article. It would be better to describe briefly in the discussion section and all figures related will move to the supplementary section.

6. (minor point) in page7, H3K37me3 should be corrected to H3K27me3.

Version 1:

Reviewer comments:

Reviewer #1

(Remarks to the Author)

I appreciate the authors response and perspectives regarding their data. However, my major concerns remain unaddressed. The authors state that their objective was to present a uniform framework for the analysis of CUT&Tag data and that one of the key benefits of CUT&Tag is the need for a reduced number of sequencing reads as compared with ChIP-seq. I think that establishing a standardized workflow is very important for the community. Thus, I think the use of a >99% duplicated library sequenced to ~135M reads is not very generalizable and is potentially misleading. High quality CUT&Tag libraries typically require no more than 5M reads and have much lower duplication levels. Thus, key analysis choices such as duplicate retention, will have very different effects in the authors dataset and in more typical libraries. Simply said, I do not think that the datasets presented in this manuscript are generalizable to what the majority of users will obtain with a high quality protocol. In my lab, libraries with this degree of duplication are discarded as failed runs.

I think there is a discussion to be had about whether deep sequencing a low quality/low complexity library to a very high depth produces similar data to a high quality/high complexity library sequenced to lower depth. This may be a case of differential use of the term "complexity". We use this term to refer to a fundamental property of the library, specifying the total number of unique DNA molecules contained. Deeper sequencing of both complex and non complex libraries can identify additional unique fragments, but does not change the underlying properties of the library. Low complexity is typically indicative of poor antibody binding, Tn5 recruitment, Tn5 activity or DNA loss somewhere in the protocol. My concern is that this is non-uniform and thus not comparable to a high complexity library. Further, over amplification of a small starting DNA pool will result in PCR biasing and further confusing interpretation of subsequent sequencing results.

I believe that this low complexity comes from the fact that H3K27Ac is a challenging mark to profile by CUT&Tag in K562 cells (I erroneously stated that the Hennikoff data had high complexity in my initial review, when in fact these data are low complexity). I also wonder if methodological optimization could improve these results. I think this concern can only be addressed through the generation of new data either with an optimized protocol and/or alternate cell line used in Encode.

In addition, the use of IgG controls is very important as it will control for any non-antibody targeted tagmentation which tends to occur at sites of chromatin accessibility that frequently co-occur with H3K27Ac.

Reviewer #2

(Remarks to the Author)

I am satisfied with the authors' responses, and I support publication.

Reviewer #3

(Remarks to the Author)

The authors addressed all questions raised by reviewers and thus modified version of manuscript is now further improved. I have no objection to publishing this article in Nat. Comm.

Version 2:

Reviewer comments:

Reviewer #2

(Remarks to the Author)

When we introduced CUT&Tag, we included benchmarking against ENCODE ChIP-seq. We did not use the ChIP-seq as the "standard". Rather we compared the datasets on equal footing, which is most useful for people considering using one method rather than another. When the authors say in the abstract "However thorough evaluation and benchmarking against established ChIP-seq datasets are still lacking" they imply that our own benchmarking tests in Kaya-Okur et al. Nat Comm 2019 were not thorough. This is opinion is inaccurate: The authors' benchmarking against ENCODE ChIP-seq considered only two histone marks for comparison, and only one antibody in common with ENCODE ChIP-seq for each, whereas we compared four histone marks to ENCODE ChIP-seq data (H3K27me3, H3K4me1, H3K4me2 and H3K4me3) and in all cases CUT&Tag provided better data. We did not test H3K27ac, but it's not surprising that CUT&Tag performance wasn't quite as good as that for ChIP-seq given that the antibody used in the ENCODE project, Abcam ab4729, is a ChIP-grade antibody, which stacked the cards against CUT&Tag. The authors consider CUT&Tag to be a "promising new method" that needs to be benchmarked against the current standard, and when we published the method in 2019 that was indeed the case. However, in mid-2023 the situation is very different, as CUT&Tag is now the unquestioned standard for low cell numbers and single cells.

Rather than being a benchmarking study, most of this manuscript is about optimisations, for example testing different post-tagmentation procedures (which we did in Kaya-Okur et al. Nat Protocols 2020). The gist of the Reviewer's criticism is that one cannot trust optimisations when the data quality is subpar, and I agree, and it's critical for optimisations to be based on data typical of that generated by others in the community. Library complexity together with signal-to-noise is critical, and Supplementary Figure 1 shows that their duplication rates were consistently much higher than ours. Therefore, I performed a Fraction of Reads in Peaks (FRiP) comparison but now including the author's own H3K27me3 data, using their best dataset based on Supplementary Figure 1 (GSE5976154, red curves in the attached figure). While the number of peaks is similar, the FRiPs are somewhat lower, but this is their best dataset, compared to a cell number series down to 200 cells.

Although the authors did a thorough job of testing different H3K27ac antibodies, now the major antibody companies are doing the same for CUT&RUN/Tag with at least histone modifications as they have in the past for ChIP-seq. As for CUT&Tag recovering at most half of the ChIP-seq peaks, this says more about peak callers than data quality, and who knows what the authors would have found if they had tested ChIP-seq against CUT&Tag peaks? Not that I'm suggesting doing that but rather to use annotated landmarks that are not biased by setting one method as a standard with the goal for the other being to capture all the peaks. As we don't know which of the ENCODE peaks are of functional significance, it seems possible that the ones found by both methods are the ones that users should care about. Instead, the authors could do their analysis by asking how well H3K27ac captures ENCODE's candidate cis-regulatory elements (cCREs), which is based mostly on DNaseI hypersensitivity data collected by the project. Or they could use annotations based on function, such as STARR-seq (PMID: 33292397) for enhancers and SuRE (PMID: 28024146) for promoters, available for K562 cells.

My FRiP analysis also confirms that the ChIP-seq FRiPs using GSM788088 from ENCODE using CST9733 (purple curves) are much lower than CUT&Tag, both for our 2019 data and for the authors' data when all are down-sampled similarly. Here I'm comparing cell numbers down to 60 cells, and even though ENCODE ChIP-seq used 1 million cells, our low-cell number data are cleaner. So setting ChIP-seq as the standard, and strongly implying with a declarative statement in the title that somehow CUT&Tag doesn't perform as well is misleading and ignores what researchers who are trying to decide which method to use really care about.

Steve Henikoff
HHMI and Basic Sciences Division
Fred Hutchinson Cancer Center

Version 3:

Reviewer comments:

Reviewer #2

(Remarks to the Author)

I appreciate the care and effort that the authors have put into the revision to address my critique, and I support publication.

(Remarks on code availability)

Point by point responses to reviewers

Reviewer #1

Remarks to the Author

Hu and Abbasova et al present a thorough comparison of CUT&Tag to ChIP-seq from the ENCODE consortium reaching several major conclusions. 1) CUT&Tag can identify up to 50% of peaks present in ENCODE, 2) That SECAR has superior operating characteristics to MACS2, 3) that duplicates should be omitted.

This sort of comprehensive optimization and validation work is an important resource for the community. Indeed, there is not a standardized workflow for CUT&Tag or quality metrics (as discussed below), which makes the comparison of datasets generated in different labs challenging. I appreciate the authors efforts to present the optimization work they have done and to provide a standardized workflow.

However, my major concern is that the complexity of the H3K27Ac CUT&Tag libraries appears to be quite low with duplication rates frequently being in the high 90% range (Supplementary Table 1, Figure 6a).

Thank you for your important feedback. While our duplication rates are high for H3K27ac CUT&Tag, we find that the main results are the same for Henikoff's data with lower duplication rates (e.g. 14.87% for H3K27ac Kaya-Okur SRR8383508 (C&T); 1.36% for H3K27me3 Kaya-Okur SRR11074239 (C&T) - **Supplementary Table 1**). Despite these high duplication rates, we were able to attain a large number of unique fragments for nearly all samples (**Supplementary Table 1**), as we sequenced our libraries deeply. For example our Abcam-ab4729 1:100 15 PCR SDS sample has 4,123,523 unique fragments despite a 96.94% duplication rate. This is comparable to Henikoff's high quality H3K27me3 data, such as H3K27me3 Kaya-Okur SRR11074239 (C&T) with 3,996,319 unique fragments and 1.36% duplication rate. Therefore, we find that ENCODE benchmarking with our data is comparable to Henikoff's high quality datasets with low duplication rates and we reach similar conclusions: CUT&Tag ENCODE coverage ceiling, when precision is balanced to >80%, is still 50% for H3K27 marks.

We also tested ENCODE capture using an aggregated CUT&Tag sample, comprising merged de-duplicated reads from each of our samples and published data from Kaya-Okur et al. (2019). This should represent the highest complexity library obtainable with our data. At >80% precision, the combined sample achieved 59.6% and 64.2.% ENCODE recall when combining 5 samples from our own experiments and additionally including the two H3K27ac samples from Kaya-Okur et al. This shows that higher recall can be achieved with combined samples. However, some of this improvement is likely to result from using a combination

of antibodies across multiple experiments, which is not really a fair comparison when evaluating the performance of a single sample (generated from a single antibody).

In the DNA isolation experiments, duplication rates appear lower (50-70% range), but it does not appear that these experiments were the ones stringently evaluated in the prior figures.

Thank you for observing and raising this point. We find that while duplication rates are lower with reduced PCR cycles, these samples have fewer unique fragments and therefore reduced library complexity. This has led us to restructure the text and figures to clarify. Specifically we have added ENCODE capture and precision plots for DNA extraction methods and PCR cycles (originally **Fig. 6**) to **Fig. 2**. This motivates why we chose the condition '15 PCR cycles and SDS-based DNA extraction' for in-depth analysis for the remainder of the manuscript. The reason being that despite these samples having a large percentage of duplicates, they have the greatest number of total unique fragments, which we were able to detect by sequencing deeply. The difference between column versus SDS-based DNA extraction could be due to a larger amount of DNA being recovered by SDS extraction, which limits sample transfers, rather than potentially losing material from columns. We hope that the restructuring which puts the experimental optimization in the beginning of the results section will improve the overall flow and understanding of the manuscript.

In our experience, high quality CUT&Tag libraries for H3K27ac (using ab4729) will have duplication rates in the range of 50-70%, but will often times be much lower and rates of <20% for H3K27me3 are quite common. As a result, the conclusions drawn in this study may not be comparable to what others will find in more complex datasets. The consequences of retaining duplicates in this dataset as compared with a dataset with a 40% duplication rate will be very different.

Even though our H3K27ac CUT&Tag libraries have high duplication rates, we attain an adequate/high number of total unique fragments (comparable to Henikoff's H3K27me3 data; **Supplementary Table 1**) due to high sequencing coverage in our data. When downsampled, our duplication rates are lower, but this results in fewer unique fragments (**Supplementary Fig. 1e**). Our main conclusions include Henikoff's data where we attain similar results even though Henikoff has low duplication rates (e.g. for H3K27me3 Kaya-Okur SRR11074239 duplication rate is 1.36%). We would like to thank you for your (later) suggestion to look into another CUT&Tag dataset in another ENCODE cell line: we assessed Henikoff's H3K27me3 data in HCT116 cells (Bartlett et al., 2021; **Supplementary Fig. 4f**) and arrived at the same conclusions for this mark.

An explanation for the low complexity of these libraries isn't completely clear. We have had good luck with H3K27Ac profiling in leukemia cell lines using ab4729 at 1:50 and the CUT&Tag@home protocol without digitonin, and routinely get duplication rates of 50%. We have found that using LowBind PCR tubes, minimizing pipette mixing or inversion of the tubes leads to maximal bead/cell retention and higher complexity libraries. Some have reported on the protocols.io comments section for the CUT&Tag protocol, that certain H3K27Ac antibodies work better in some cell lines than in others. Indeed, the Kaya-Okur CUT&Tag for H3K27Ac in K562 cells does not have exceedingly high signal to noise (based on our groups

internal analysis of this data), so this may be a cell line specific issue. We have not evaluated K562s but have had reasonable luck with most leukemia cell lines and primary hematopoietic cells we have tried.

This is very valuable advice, which would be helpful for the community. We reach the same conclusions with Henikoff's data using H3K27me3 in K562 cells. Even though H3K27ac may not be ideally profiled in K562, we selected this due to the availability of K562 ENCODE data for benchmarking purposes. Comprehensive benchmarking of other ENCODE cell lines would be outside the scope of this study, but represents a helpful future direction. However, we have included novel analyses of Henikoff's H3K27me3 data in HCT116 cells (Bartlett et al., 2021; **Supplementary Fig. 4f**) and arrived at the same conclusions for this mark. We used this study as an opportunity to identify suitable analysis approaches based on ENCODE benchmarking of high quality data from Henikoff for H3K27me3, as well as our data, where altogether we arrived at similar conclusions. We provide a framework for CUT&Tag experimental design and analysis to help the user select the optimal antibody for their cell line/tissue of interest.

Major points

1. Comparison of CUT&Tag with Encode

a. As the authors performed CUT&Tag for only two histone marks and ENCODE covered many histone marks in addition to transcription factors, it would probably be better to have a more specific title (i.e. ENCODE H3K27Ac and H3K27me3 peaks) and conclusions as this work did not cover all of ENCODE.

This is a valid point and we have changed the title to more appropriately reflect the scope of the study to "CUT&Tag recovers up to half of ENCODE CHIP-seq peaks in modifications of H3K27"

b. The low complexity of the libraries precludes making definitive statements about whether CUT&Tag can satisfactorily capture ENCODE peaks.

Please see points above regarding our recovery of a comparable number of unique fragments to Henikoff's data with low duplication rates. We also tested precision and recall in two aggregate samples (combining all 5 samples from this study and the 5 samples plus two from Henikoff), and reach similar, albeit slightly higher ceilings of ENCODE recovery (~60%).

c. With high complexity libraries we frequently sequence deeper (10M reads) and retain duplicates which results in higher peak numbers and improves power in differential binding experiments. It would be interesting to see if higher complexity libraries sequenced deeper would capture a greater percentage of ENCODE peaks.

Because we have sequenced very deeply (up to 135M mapped fragments, see **Supplementary Table 1**), we were able to attain high complexity libraries, despite higher duplication rates. We find that retention of duplicates gives high capture, but at the expense of precision, i.e. presence of new (spurious) peaks that do not make sense in chromatin annotation (e.g. **Supplementary Fig. 5c** - duplicate-only peaks

ChromHMM-annotated to heterochromatin for H3K27ac). It is difficult to establish the ground truth for chromatin profiling. The approach we take in the analysis is to balance precision and recall as done previously for CUT&RUN (Meers et al., 2019), but has not yet been reported in the literature for CUT&Tag. In our samples with lower duplication rates e.g. Active Motif 39133 1:100 11 PCR column with a duplication rate of 46.33%, we get low unique fragments: 786,081 (see **Supplementary Table 1**). This translates into poor precision and recall despite the reduction in duplication rates, e.g. as compared to Abcam-ab4729 1:100 15 PCR SDS with 96.94% duplication rate, but 4,123,523 unique fragments and greater ENCODE capture and precision. As described in our answer to a previous question, to achieve a more complex library by an alternative approach, we have aggregated our five K562 samples across antibodies and still found a ceiling of <60% recall even under these conditions of artificially increased complexity.

d. Perhaps the authors could repeat these studies on another cell line used in ENCODE, generate higher complexity data and generate more generalizable conclusions?

We repeated our full analysis pipeline in another ENCODE cell line with published CUT&Tag data: H3K27me3 in HCT116 cells (Bartlett et al., 2021). We find that our main conclusions still hold (**Supplementary Fig. 4f**). This is likely due to the fact that we used Henikoff's high quality H3K27me3 data to arrive at our main conclusions. Because we sequenced deeply, we were able to attain comparable unique fragments to Henikoff despite our high duplication rates. Although it would be interesting to study a wide range of cell types, we were limited by the cell lines available in ENCODE. K562 is a common cell line used in epigenomic methods development (e.g. Kaya-Okur et al., 2019; Wu et al., 2021; Janssens et al., 2022), so we thought it would be helpful to use this particular ENCODE cell line for benchmarking. We addressed complexity using our 'pseudo-bulked' aggregate sample with additive complexity from all datasets assessed per histone mark.

2. Comparison of SEACR to MACS2

- a. Another issue that is not comprehensively addressed is that the peaks produced by SEACR are very wide and promoter peaks frequently are merged with nearby enhancers in our experience. This can have significant downstream ramifications, depending on goals of secondary analysis. It is often preferable to have narrower peaks when performing differential peak calling and motif enrichment studies. In these instances, a peak caller that calls narrower peaks may be more appropriate.*
- b. MACS2 may perform better on higher complexity datasets.*
- c. Other peak callers have been developed (as cited by the authors) which have not been benchmarked here.*

These are very important and pertinent points to look into. As MACS2 and SEACR are the main peak callers used for CUT&Tag, we decided to focus on these two for benchmarking. Moving forward it would be interesting to benchmark additional peak callers, especially those specifically designed for CUT&Tag like GoPeaks.

We plotted MACS2 versus SEACR peak widths, as well as those overlapping with promoter peaks to confirm that H3K27ac SEACR peaks are wider than MACS2 peaks but find less of a difference for H3K27me3 (**Supplementary Fig. 3g**). We find that MACS2 calls a lot of narrow peaks that are not replicable (poor precision), and that SEACR gives higher precision than MACS2 (**Fig. 4b**).

The co-occurrence of K562 DNase-seq-overlapping STARR-seq enhancers with promoter region-overlapping SEACR and MACS2 peaks was also calculated to find slightly higher enhancer incidence amongst SEACR peaks, which was more pronounced - but still low - when correcting for total sample peak numbers (**Supplementary Fig. 3h**). We also calculated the average number of sample peaks falling within promoter regions to find ratios of ~1 for both SEACR and MACS2. Since the values for MACS2 peaks were somewhat higher, taken together with reduced capture of putative K562 enhancers, this suggests that there are cases where MACS2 discriminates between nearby promoters and enhancers and SEACR does not. Nevertheless, the difference between SEACR and MACS2 is not substantial.

For TFBM analysis, we used constant 1000bp size parameter in HOMER (500-1000bp recommended for histone marks) - i.e. searching 1000bp region centered around the peak summit, so this would be consistent for both SEACR and MACS2 for motif enrichment studies. Consequently, any differences in peak width will matter less. From this we find that both peak callers detect some motifs that are not in ENCODE.

With regard to differential peak analysis we would raise the important point that many of the shorter and smaller peaks identified by MACS2 (in CHIP-seq data) show regional correlations. In that sense, there may actually be an increase in power to identify differential acetylation (or other modifications), when broader peaks are called and small peaks grouped together. Such correlations were for example observed in our previous work, profiling H3K27ac in human post-mortem brain samples from Alzheimer's disease patients and controls (Marzi et al., 2018).

3. Retention of duplicates

a. As described above, this will be highly dependent on the degree of duplication and more complex libraries will be less impacted by duplication. In this setting, retention of duplicates may be appropriate.

We are trying to establish a consensus framework for CUT&Tag analysis that would be suitable for a range of data types so that CUT&Tag datasets from different groups can be comparable. We find that keeping duplicates increases ENCODE capture slightly at the cost of poor precision, so it does not appear to be worth it. We found spurious peaks upon retention of duplicates especially with MACS2. If duplicates were real, they should map onto non-spurious sites – this is not the case for both ours and Henikoff's data.

We agree that if duplication rates are low, there is minimal benefit in including them (**Fig. 5b**) both in peak calling and differential analysis. Furthermore we find that precision is reduced with retention of duplicates, e.g. picking up heterochromatin for H3K27ac (**Supplementary Fig. 5c**). This is particularly relevant to MACS2 peak calling, where the number of duplicate-only peaks was substantial.

Minor Points

- The discussion about the brain in the introduction seems out of place considering all the data presented are in a leukemia cell line*

We have adjusted this in the introduction to encompass complex human disease in general and refer to brain disease only once as an example.
- The fonts in the figures are frequently very small and difficult to read*

We have adjusted the fonts on the figures.
- More closeup images of tracks and called peaks would enable readers to better assess data quality*

We have added examples of close up tracks in **Supplementary Fig. 3f** and provided URLs of track ranges showing MACS2 and SEACR peak calling, so that the reader can examine these peaks in detail with their preferred settings: “Sample bigwig tracks can be found at: http://genome-euro.ucsc.edu/s/bschilder/cutntag_benchmarking”
- Listing peak number in addition to percentage overlap with Encode would be valuable*

We have listed all peak numbers in **Supplementary Table 2**. We find that adding this information to our main figures would appear crowded.
- Please include the ENCODE ID in the text and methods.*

This information has been added to the methods. We found inclusion in the main text to be overcrowded, but could include it there as well in a future version, should the editors decide it is needed in the main text.
- Please include the source of the ATAC-Seq dataset. Is it from ENCODE too?*

This information has been added to methods: “ATAC-seq libraries (ENCODE IDs: ENCLB918NXF and ENCLB758GEG) used for FRiP analysis were processed with the nf-core ATAC-seq pipeline ⁶¹.”

- *For SEACR peak calling, did you used any control samples like IgG?*
- *For MACS2 peak calling, did you used any control samples like IgG?*

We have included Henikoff’s IgG sample in our analysis and it did not make a difference (**Supplementary Fig. 3e**). We did process our own IgG sample, but it was processed on a separate date, so it may not be comparable. Additionally, with IgG, our library yield is between 0.1-0.2 ng/μl using 15 PCR cycles and either DNA extraction method. This is barely enough for sequencing. We find that there is sufficient yield for sequencing the IgG control in TIP-seq (CUT&Tag with linear amplification; Bartlett et al., 2021), so this could be a robust way to subtract out background noise in spurious transposase activity moving forward.

- *Figure 7a and 7b are blurry*

We fixed these two figures.

- *What is the dashed line in Figure 4G?*

The dashed line represents the ENCODE samples. This has been explicitly clarified in the Figure legend (now **Fig. 5g**).

- *Line 435: Please include number of peaks fed into HOMER motif analysis for each method to highlight any differences/similarities since that can really affect the magnitudes of HOMER p-values.*

All peaks were used for each sample. Peak information, including total peak numbers, can be found in **Supplementary Table 2**.

Theodore P. Braun MD, PhD
 Assistant Professor
 Division of Hematology & Medical Oncology
 Division of Oncologic Sciences
 Knight Cancer Institute
 Oregon Health & Science University

Reviewer #2

Remarks to the Author

The authors optimize CUT&Tag and benchmark it against ChIP-seq. In light of the rapidly growing popularity of CUT&Tag for low cell numbers and single-cells, this study is very timely. Also, there is a lot of useful information that users will find helpful, such as what antibody concentrations to use and showing that consistent results are obtained for an acetyl mark in the presence or absence of HDAC inhibitors. Although I have no criticisms of the experimental aspects of the work or the functional annotations, I found flaws in the analysis that undermine the major conclusion that is the title of the paper. The comparisons

that the authors do are based on peak calling, but their analysis did not sufficiently compensate for differences between CUT&Tag and ChIP-seq data. There is no ground-truth set of peaks given the vagaries of peak finders - they will all find peaks in high-signal regions and different sets of noise in low-signal regions, and showing that different peak callers or parameters gives similar results doesn't solve the problem. Further analyses are needed to make this part of the manuscript suitable for publication in Nature Communications.

Major issues

1) The observation that CUT&Tag shows a higher concentration of reads over read summits than ENCODE ChIP-seq can explain why it has lower FRiP values than ChIP-seq, because ChIP-seq peaks are wider and so will capture more reads all else being equal. This is evident from Figure 4f, where the narrower CUT&Tag peaks indicate better resolution than ChIP-seq. In FRiP analysis, the tighter concentration of CUT&Tag reads over the summit is a disadvantage, and it casts doubt on the value of all the FRiP comparisons between CUT&Tag and ChIP-seq in the paper. One way to address this issue is to compare their sets of peaks in a peer-to-peer manner, for instance using the bedtools intersect tool.

Thank you for these insightful suggestions. In attempts to correct for potential differences in breadth and read capture of ENCODE and CUT&Tag peaks, FRiPs calculations were repeated using intersecting ENCODE and CUT&Tag peak regions, which were determined separately for each CUT&Tag sample (**Supplementary Fig. 2c**). This revealed that while H3K27ac FRiP scores were relatively similar for CUT&Tag and ENCODE, H3K27me3 CUT&Tag produced over twice as many reads in overlapping intervals than ENCODE. One possible interpretation is that the presence of spurious peaks in ChIP-seq data inflates the H3K27ac ENCODE FRiP scores, and these regions of spurious signal are not accompanied by higher noise levels genome-wide. Otherwise, they would be excluded by the peak calling algorithm.

2) The most obvious interpretation of the observation that CUT&Tag captures the most significant peaks but misses the less significant peaks (Fig. 4c) is that the less significant peaks are false positives as a result of noise, and this is understandable in part because of the broader ChIP-seq peaks using the same peak-caller pointed out above. In other words, the broader peaks of ChIP-seq may be broad because the overall background is higher, as previous work has shown (but in contrast to statements in the paper), and this makes it difficult for a peak-caller to distinguish true from false positives. This possibility is suggested by the graph in Figure 4f, where the CUT&Tag curves are still descending at -0.5 and +0.5 kb, whereas the ENCODE ChIP-seq curve had completely leveled off by about ± 0.2 kb. The fact that the CUT&Tag curves intersect the ChIP-seq curve at ± 0.5 kb and are still descending is clear evidence that the CUT&Tag background levels are lower genome-wide and provides support for the above explanation. At a minimum the authors need to extend the horizontal axis to the point that the CUT&Tag data levels off.

We have extended the axes to ± 1 kb (now **Fig. 5f**). The CUT&Tag and ENCODE curves intersect, but CUT&Tag average read counts do not fall below those of ENCODE within these intervals. Something similar

can be observed in **Fig. 4d**. As we are surveying narrow intervals, this does not support the assertion that H3K27ac ENCODE ChIP-seq has higher levels of noise genome-wide. Furthermore, ChIP-seq peaks are not necessarily broader than CUT&Tag peaks, although CUT&Tag read density, and distribution, appears higher and broader around peak summits.

This interpretation is also consistent with the finding of higher read density for CUT&Tag over TSSs (Figure S5e). More to the point, unless the authors can rule out this explanation, the title of their paper "CUT&Tag recovers up to half of ENCODE ChIP-seq peaks" is simply explained by the higher background of ChIP-seq. See my suggestion above to help address this issue.

Many thanks for these points. We have added this point to the discussion regarding ChIP-seq peaks missed by CUT&Tag potentially being noise or false positives due to the chromatin shearing and sonication involved in the ChIP-seq protocol: "ChIP-seq peaks missed by CUT&Tag could potentially represent noise or false signal detected by ChIP-seq due to chromatin shearing and sonication, as well as fixation and cross-linking resulting in heterochromatin bias^{11,12}."

We realize that there was some confusion and have implemented your previous suggestion to extend the plotting window (now **Fig. 5f**). However, we would caution that the distribution around peak centers and TSSs alone does not necessarily have to be representative of genome-wide noise levels. We have additionally added a URL to browseable genome-wide tracks of histone modifications profiled here, which show that background noise levels of H3K27ac CUT&Tag are equal or worse than those of ENCODE ChIP-seq: http://genome-euro.ucsc.edu/s/bschilder/cutntag_benchmarking

3) The authors state in the Methods section that they used only one end of the paired-end read data for CUT&Tag, presumably because the ChIP-seq data was only single-end. The authors must suppose that using only half of the CUT&Tag data is fair, but it is not, as the point of doing paired-end reads is to get the precise position of the fragment, whereas the ENCODE Project purposely did only single-end reads because it was cheaper and faster than paired-end sequencing and there was little justification given the low resolution of ChIP-seq. Here the authors did PE75 sequencing and used this to plot the length distribution in Fig. 2b, and so to use the data properly they should align the fragments and use that for peak calling and FRiP analysis to see how well CUT&Tag can do when performed as intended. Otherwise, they are degrading the CUT&Tag data in a way that users will not. This not only biases the result in favor of ChIP-seq, but is totally unrealistic, as researchers wouldn't think of throwing away half of the data.

This is an important point, thank you for bringing it up and giving us the chance to clarify: We did use the full dataset, i.e. paired-end reads for the analysis. We have added this as a clarifying point in the methods - data processing: "The full dataset, i.e. paired-end reads were used for the analysis." Single-end reads were only used to produce heatmaps, ensuring that we keep total read counts constant (eg. **Fig. 5a**).

4) Since the authors have paired-end data they should use it like any researcher would do. SEACR requires a track (bedgraph file) as input, but MACS2 prefers a bed file of mapped fragments (BEDPE option with PE data). It can perform model-building when given mapped fragments instead of a track In all cases,

fragments mapped to chrM should be removed before peak-finding as they will skew a model-based peak finder.

As clarified in response to question 3, we used paired-end data for all data analysis. The advice regarding reads mapping to the mitochondrial genome is very helpful. We updated **Supplementary Table 1** to include percentages of mitochondrial reads for all samples to find the vast majority to be acceptably low. Several samples with higher mitochondrial content were reprocessed to test how the presence of mitochondrial reads affects peak calling. All peaks that were called when mitochondrial reads were excluded were also detected when they were included, alongside a smaller number of additional peaks. ENCODE recall was unaffected, but the absence of mitochondrial reads slightly improved precision scores (**Supplementary Table 3**).

5) The text referring to Figure 2c is confusing. The text states that "all H3K27ac CUT&Tag FRiP scores were markedly lower than the ENCODE H3K27ac ChIP-seq score of 42%" but this contradicts the previous sentence, and in Figure 2c all of the histogram bars are below 42%. If anything the opposite seems to be the case: For example using Abcam 4729 CUT&Tag peaks are ~35% whereas ENCODE peaks are ~20%. The wording in the text is perhaps ambiguous, in that one might conclude that the comparison is done by calling ENCODE peaks and calculating the fraction of ENCODE and CUT&Tag reads in the ENCODE peaks, which would be trivial, like saying a house is more like itself than it is like the house across the street. Regardless, there needs to be some clarification, since FRiP comparison is how data quality is judged, and the issue raised by the title of this article suggests that there is a data quality issue that the study tries to resolve. FRiPs should be used with extreme caution when comparing different peak finders because just producing more peaks or wider peaks regardless of quality will result in a higher FRiP value.

Thank you for allowing us to clarify this aspect of the text. 42% is referring to FRiPs of ENCODE reads in ENCODE peaks (reported by ENCODE) and we were comparing this to FRiPs of CUT&Tag reads in CUT&Tag peaks. We added this clarifying point to the text: "However, all H3K27ac CUT&Tag FRiP scores were markedly lower than ENCODE H3K27ac ChIP-seq's own reported score of 42%." (now **Fig. 3b**).

The comparison done was indeed by calling ENCODE peaks and calculating the fraction of ENCODE and CUT&Tag reads in ENCODE peaks. The purpose of doing this was to see which CUT&Tag antibody yields the most reads in ENCODE peaks. We agree that FRiPs should be used with extreme caution when comparing different peak callers. This is why we showed the FRiPs results with different peak callers, as well as with inclusion versus exclusion of duplicates (now **Fig. 3b**) to highlight the variability of FRiPs depending on the analysis approach.

Minor issues

I found misleading the use of the term "gold standard" in the Abstract and elsewhere to be somewhat misleading. This term is generally understood as closest to ground truth: "the best, most reliable, or most

prestigious thing of its type." But setting up ENCODE as the gold standard is prejudicial, because only after the results of the comparison are evaluated can it be decided which if either should be considered the gold standard, and even then it's tricky to make a call. Calling ENCODE ChIP-seq "a de facto gold standard" or "the current gold standard" is better, but just "current standard" would be more accurate.

We completely agree with this point. We have edited the manuscript text everywhere to refer to ENCODE as a "(current standard) reference panel", dropping the use of "gold standard".

The authors need to state whether or not the antibodies used were the same between ENCODE and CUT&Tag. Checking GenBank shows that this was the case for both H3K27ac (Abcam ab4729) and H3K27me3 (CST 9733), which is good. However, whereas the H3K27me3 antibody is monoclonal, the Abcam H3K27me3 antibody is polyclonal, and this complicates a direct comparison. Since multiple antibodies were used for H3K27ac and ab2729 was one of the best, this is unlikely to be a major issue, but this caveat should be mentioned.

We agree that this is an important piece of information to mention and discuss in the manuscript. We have added the information in the text, highlighting that H3K27ac antibody Abcam-ab4729 was the same one used in ENCODE, as was H3K27me3 antibody Cell Signaling Technology-9733. Interestingly, in the overall evaluation the Abcam polyclonal antibody ab4729 performed better than the monoclonal (ab177178) for H3K27ac. We added to the discussion the possibility that this could be due to the fact that Abcam ab4729 was the same antibody used in ENCODE for H3K27ac: "Because our best performing antibody in H3K27ac CUT&Tag is the same as that used for ENCODE ChIP-seq (Abcam 4729), this could have contributed to its favorable result in ENCODE overlap and comparisons."

Reviewer #3

Remarks to the Author

This article showed the performance of CUT&Tag and comparison with ENCODE ChIP-seq. The epigenomic profiles like histone modifications can be interpreted by dissecting DNA-protein interactions using next generation sequencing. The authors generated their own 30 datasets by the CUT&Tag method with multiple commercially popular antibodies against H3K27ac and H3k27me3. They performed a massive computational analysis to benchmark CUT&Tag data against ChIP-seq data from ENCODE. The result is mostly consistent with the previous notion; CUT&Tag recovers no more than half of ChIP-seq peaks. They tested library preparation method (PCR cycles and DNA purification method), compared two peak callers with different parameters, and checked precision and recall rate to optimized the best condition.

Overall, the idea and flow of this article are quite informative and invoke interest for ones seeking the epigenomic features. However, the original CUT&Tag technique was published in 2019 and afterwards

multiple papers were used this method to show different applications. The analysis pipelines like SEACR and MACS2 were developed by other groups. The only progress is a systematic comparative analysis of two analysis pipelines using CUT&Tag and ChIP-seq datasets. Therefore, I think the important advances of significance to epigenetic specialist seem to be relatively weak. The authors did not address quite well the question why CUT&Tag could not recover the true ChIP-seq peaks. Instead they emphasized only the benchmarking result and optimization.

Thank you for your comprehensive and nuanced summary. You raise excellent points. We have not seen limited recovery of established ChIP-seq signals (such as ENCODE) by CUT&Tag reported previously and do believe this notion is novel and useful for the scientific community. We agree that SEACR and MACS2 were developed by other groups. Our aim here is to systematically benchmark CUT&Tag against current standard ENCODE for H3K27 marks to demonstrate CUT&Tag performance quantitatively to the broader field. This will hopefully help onboard more users to this method to address their low-input epigenomic profiling questions. Because CUT&Tag is a relatively new method compared to ChIP-seq that has been around for decades, we deemed it useful to have a framework for experimental design and analysis that non-experts in epigenomic analysis and experts alike can refer to. We propose a standard framework for CUT&Tag experimental design and analysis, which is important to have due to e.g. presence of highly variable QC FRiP scores depending on the analysis method used (now **Fig. 3b**). We highlight the importance of balancing precision and recall in CUT&Tag data to appropriately benchmark and facilitate selection of the optimal antibody and experimental conditions. We functionally characterize the peaks missed by CUT&Tag. In the absence of orthogonal approaches to validate ChIP-seq data, we can hypothesize the biological relevance of ChIP-seq peaks missed by CUT&Tag based on whether their chromatin annotation makes sense with respect to the histone mark.

The benchmarking result and optimisation was the main purpose of this paper. A similar systematic benchmarking approach has not been previously reported for CUT&Tag studies. Validating a method against current standard ChIP-seq is important to allow us to understand how we can suitably analyze and interpret CUT&Tag data, as well as to optimize its performance to maximize its utility for low input and single cell applications.

Major issues

There are many points to be considered.

1. The nature of H3K27ac and H3K27me3 might not be properly appreciated; H3K27ac shows generally narrow point peak profiles and H3K27me3 broad peak patterns. Direct comparison of these two histone modification profiles is expected to be incompatible. As you know, the histone modification of H3K4me3 is usually enriched at the promoters and show the similar narrow peak shapes like that of H3K27ac. Rather than H3K27me3, H3K4me3 would give a better result to support your benchmarking outcome of H3K27ac CUT&Tag

Thank you for raising these important points and giving us the chance to clarify: Our aim was to benchmark each histone mark against ENCODE, not necessarily to each other. Overall, we chose 1 active mark and 1 repressive mark each as an initial sampling of both types of histone modifications. Specifically, we chose H3K27me3, as it is the positive control recommended by Henikoff (and therefore widely adopted in the field) and expected to give high quality CUT&Tag data. Thus we included Henikoff's H3K27me3 CUT&Tag data in our benchmarking. We optimized for H3K27ac, as it is a currently underexplored histone mark in CUT&Tag, yet plays a critical role in epigenetic mechanisms of complex human disease (e.g. Marzi et al., 2018, Nott et al., 2019). Even in the presence of different natural peak widths of these two histone marks, our results still point to the superior performance of SEACR (which calls wider peaks) for both histone marks.

2. The enzymatic activity and efficiency of Tn5 affect the data quality for CUT&Tag which is principally different from ChIP-seq where the antibodies bound to target proteins are pulled down. That's why the discrepancy between CUT&Tag and ChIP-seq appears. I think it would be better to interpret you data as a complementary cooperation between two methods to interpret the biologically natural epigenetic modifications.

This is a great point, which we tried to address in the following ways: Chromatin annotations of ChIP-seq peaks missed by CUT&Tag provide insights into the biological relevance of these ChIP-seq peaks. *In situ* tagmentation of antibody-bound sites in CUT&Tag could potentially lead to lower background noise as compared to chromatin cross-linking, fragmentation and sonication before antibody binding in ChIP-seq (Rodriguez-Ubeva & Ballestar, 2013, Teytelman et al., 2013). CUT&Tag can be more easily adapted to a range of single cell applications in both plate/microwell based (e.g. Takara iCELL8), as well as droplet microfluidic-based systems (e.g. 10x, Drop-seq). We discuss the potential limitations of ChIP-seq in the introduction to motivate the use of CUT&Tag. However, upon evaluation of CUT&Tag, we find that it has limitations as well, such as limited recovery of ENCODE peaks for H3K27 marks, including those mapping to active regulatory elements and of putative importance; efficiency loss from the transposase inserting adapters in the correct orientation to amplify half of the time; and PCR amplification biases from PCR-based library preparation where errors are exponentially propagated.

3. The figures should be reorganized. For example, Fig 1 can be moved to supplementary section because the experimental methodology and analysis pipelines are not original and just help readers for easy understanding your analysis. Fig 3a and Supp. Fig2c are overlapped. And many parts in main Figures are partly overlapped with the supplementary figures, which are redundant and need to be simplified.

Thank you for providing feedback on the figures. Regarding **Fig. 1**, we believe it can act as a useful overall summary for the reader, as it covers the entire experimental design and analysis framework we present. We would leave the decision regarding whether to include **Fig. 1** in the main results section or move it to Supplementary material to the editors. It is correct that some of the content in the supplementary figures mirrors parts seen in the main text. The supplementary figures show all samples side by side (including

those that were deemed to make the main figures overly crowded) to allow for easier visual comparison. We have edited descriptions to better highlight which samples were further added in the Supplementary figures. We believe that this is a regular use case of supplementary figures. We have also reshuffled the figures and text, moving the experimental optimization Figures 5 and 6 to Figure 2 to rationalize the samples we selected for in depth analysis in the remaining figures and improve the flow of the manuscript.

4. In Fig.4b, there is no data for with and without duplicates.

We've now added this in.

5. The section, "Experimental optimisation of CUT&Tag" is a minor point in this article. It would be better to describe briefly in the discussion section and all figures related will move to the supplementary section.

We have moved the section regarding experimental optimization, including PCR cycles and different DNA extraction methods to an earlier part of the paper to provide rationale for selecting specific samples for our in depth benchmarking. We believe the question around addition of HDAC inhibitors in native CUT&Tag conditions for H3K27ac will be of interest to researchers and helpful to note that it does not make a difference. We have consolidated the experimental optimization panels into one figure and hope that the updated structure and condensed figure will provide better understanding and flow to the manuscript.

Minor issues

6. (minor point) in page7, H3K37me3 should be corrected to H3K27me3.

We fixed this, thank you for spotting it.

Responses to reviewer 1's comments:

I appreciate the authors response and perspectives regarding their data. However, my major concerns remain unaddressed. The authors state that their objective was to present a uniform framework for the analysis of CUT&Tag data and that one of the key benefits of CUT&Tag is the need for a reduced number of sequencing reads as compared with ChIP-seq. I think that establishing a standardized workflow is very important for the community. Thus, I think the use of a >99% duplicated library sequenced to ~135M reads is not very generalizable and is potentially misleading. High quality CUT&Tag libraries typically require no more than 5M reads and have much lower duplication levels. Thus, key analysis choices such as duplicate retention, will have very different effects in the authors dataset and in more typical libraries. Simply said, I do not think that the datasets presented in this manuscript are generalizable to what the majority of users will obtain with a high quality protocol. In my lab, libraries with this degree of duplication are discarded as failed runs.

Only one sample (out of six) was sequenced to 135M reads with 97% duplication. This sample did not drive the main conclusions of our manuscript. However, we would be amenable to removing this sample from the manuscript, should the editors feel this is necessary. Importantly, in light of similar concerns about high duplication rates and potentially unrepresentative samples, we also used published CUT&Tag datasets from Kaya-Okur et al. and reached similar conclusions. Of note, the data produced by the Henikoff lab was generated in a high-quality cell line sample (K562 cells), with an assay that was developed and optimized by this group. The fact that their H3K27ac CUT&Tag libraries achieve similar library complexity and benchmarking results as our data, suggests that this may be the ceiling of what is achievable in this context.

I think there is a discussion to be had about whether deep sequencing a low quality/low complexity library to a very high depth produces similar data to a high quality/high complexity library sequenced to lower depth. This may be a case of differential use of the term "complexity". We use this term to refer to a fundamental property of the library, specifying the total number of unique DNA molecules contained. Deeper sequencing of both complex and non complex libraries can identify additional unique fragments, but does not change the underlying properties of the library. Low complexity is typically indicative of poor antibody binding, Tn5 recruitment, Tn5 activity or DNA loss somewhere in the protocol. My concern is that this is non-uniform and thus not comparable to a high complexity library. Further, over amplification of a small starting DNA pool will result in PCR biasing and further confusing interpretation of subsequent sequencing results.

The reviewer presents a discussion on library complexity as a fundamental property of a sequencing library. However, this definition is not practical, because complexity can only be determined by sequencing the library. It is of note that we used PCR cycle numbers equal to or lower than those reported in the protocol by the Henikoff lab. In this regard, we also undertook multiple optimisations and sensitivity analyses, including varying the PCR cycle number, downsampling our fastq files to 8M paired end reads, analysing our data with and without retention of duplicates none of which improved our overall results (specifically

ENCODE precision and recall). These results hold up across all five main in-house histone acetylation CUT&Tag experiments and the two samples published by Henikoff and colleagues and therefore show a degree of generalisability across antibodies and laboratories. Of note, to additionally address the reviewer's previous concerns we generated in silico merged high-complexity libraries, which the reviewer seems to have missed. This sample showed a modest improvement in ENCODE capture, which we now remark on in the discussion:

"Analysis of both current and comparison data showed that CUT&Tag achieves a maximum of ~50% ENCODE coverage for the H3K27ac mark, and capture was modestly improved to ~60% when using an aggregate high-complexity sample. This suggests joint peak calling across multiple merged samples may improve library complexity and signal capture."

I believe that this low complexity comes from the fact that H3K27Ac is a challenging mark to profile by CUT&Tag in K562 cells (I erroneously stated that the Hennikoff data had high complexity in my initial review, when in fact these data are low complexity). I also wonder if methodological optimization could improve these results. I think this concern can only be addressed through the generation of new data either with an optimized protocol and/or alternate cell line used in Encode.

The reviewer points out that H3K27ac may be a challenging mark to profile in K562 cells. To this point a) the CUT&Tag assay should work across many contexts and not depend on the underlying cell line/tissue type, b) if it is challenging to use the assay in a high-quality cell line, then it is likely to be even more so in lower quality biological samples, eg. postmortem brain tissue, tumour samples, or any sample having undergone perturbations or extensive storage, c) the message of the paper is that practically the assay does not perform as well as suggested across all marks and biological samples. While the reviewer argues that it may work better in specific contexts, we would argue that the method should be largely independent of context, such that the questions that researchers might choose to interrogate using the assay are not drastically limited. Importantly and to specifically address this reviewer's concerns in the first round of revisions we added data on an additional published cell line: ENCODE capture was evaluated for H3K27me3 in HCT116 cells (Bartlett et al., 2021; **Supplementary Fig. 4f**) and we arrived at the same conclusions for this mark.

In addition, the use of IgG controls is very important as it will control for any non-antibody targeted tagmentation which tends to occur at sites of chromatin accessibility that frequently co-occur with H3K27Ac.

To address this point in our revised manuscript we had added an evaluation of peak calling approaches including negative control IgG samples generated by Henikoff and colleagues. This did not improve our overall results as evaluated by ENCODE precision and recall. To more directly address this point we have now recalled the published peaks reported by Henikoff and colleagues (Kaya-Okur et al., 2019) with the IgG control samples generated from the same samples in the same lab. We have now added these results to **Supplementary Fig. 3e**, reconfirming that the use of IgG controls does not improve ENCODE capture.

Response to Reviewers - NCOMMS-22-30416B-Z “CUT&Tag recovers up to half of ENCODE ChIP-seq peaks in modifications of H3K27”

We would like to thank Professor Henikoff for his review of our paper “CUT&Tag recovers up to half of ENCODE ChIP-seq in modifications of H3K27”. We are huge fans of the methods he has developed and are enthusiastic users of CUT&Tag in many of our recent studies. Importantly, we intended the manuscript to be an objective comparison of CUT&Tag against ENCODE ChIP-seq rather than a criticism of the method. We have highlighted this throughout the manuscript, including in the discussion on the incomplete overlap between the techniques.

When we introduced CUT&Tag, we included benchmarking against ENCODE ChIP-seq. We did not use the ChIP-seq as the "standard". Rather we compared the datasets on equal footing, which is most useful for people considering using one method rather than another. When the authors say in the abstract "However thorough evaluation and benchmarking against established ChIP-seq datasets are still lacking" they imply that our own benchmarking tests in Kaya-Okur et al. Nat Comm 2019 were not thorough. This opinion is inaccurate: The authors' benchmarking against ENCODE ChIP-seq considered only two histone marks for comparison, and only one antibody in common with ENCODE ChIP-seq for each, whereas we compared four histone marks to ENCODE ChIP-seq data (H3K27me3, H3K4me1, H3K4me2 and H3K4me3) and in all cases CUT&Tag provided better data. We did not test H3K27ac, but it's not surprising that CUT&Tag performance wasn't quite as good as that for ChIP-seq given that the antibody used in the ENCODE project, Abcam ab4729, is a ChIP-grade antibody, which stacked the cards against CUT&Tag. The authors consider CUT&Tag to be a "promising new method" that needs to be benchmarked against the current standard, and when we published the method in 2019 that was indeed the case. However, in mid-2023 the situation is very different, as CUT&Tag is now the unquestioned standard for low cell numbers and single cells.

We agree with Professor Henikoff that CUT&Tag is the method of choice for low-input and single-cell applications and we have changed that terminology throughout the manuscript to reflect this.

We intended the manuscript to be an objective comparison of CUT&Tag against ENCODE ChIP-seq and our paper adds further dimensions to the benchmarking in Kaya-Okur et al. This includes: 1) systematically working through experimental and computational setup and parameters (including five different antibodies for H3K27ac), 2) benchmarking across multiple different metrics (read correlations, ENCODE recall and precision, STARR-seq analyses, dissection of ATAC-seq signal, peak and signal enrichment metrics, functional annotation and TF-binding motif overlap analyses), and 3) focusing particularly on H3K27ac, which as noted by the Reviewer, lacks previous benchmarking. In the context of complex human diseases, H3K27ac stands out as a particularly important histone modification. This is both because genetic risk for these

diseases has been shown to be enriched in regions of the genome marked by H3K27ac (cell-type specifically), and because the modification is altered in disease-affected tissue. This includes H3K27ac signatures identified in post-mortem brain samples in Alzheimer's disease (Marzi et al., 2018, Nativio et al., 2020, Ramamurthy et al., 2022) autism spectrum disorder (Sun et al., 2016) schizophrenia and bipolar disorder (Ghirdar et al., 2022). To highlight this, we have changed the manuscript title to focus on the acetylation data: "**CUT&Tag recovers up to half of ENCODE ChIP-seq histone acetylation peaks**".

Rather than being a benchmarking study, most of this manuscript is about optimisations, for example testing different post-tagmentation procedures (which we did in Kaya-Okur et al. Nat Protocols 2020). The gist of the Reviewer's criticism is that one cannot trust optimisations when the data quality is subpar, and I agree, and it's critical for optimisations to be based on data typical of that generated by others in the community. Library complexity together with signal-to-noise is critical, and Supplementary Figure 1 shows that their duplication rates were consistently much higher than ours. Therefore, I performed a Fraction of Reads in Peaks (FRiP) comparison but now including the author's own H3K27me3 data, using their best dataset based on Supplementary Figure 1 (GSE5976154, red curves in the attached figure). While the number of peaks is similar, the FRiPs are somewhat lower, but this is their best dataset, compared to a cell number series down to 200 cells.

To address Professor Heinokoffs comment regarding library complexity we performed a series of new experiments. Our revised manuscript includes two new replicates for each of the highest-performing antibodies for H3K27ac and H3K27me3. The duplication rates in the new dataset ranged from 11 to 33%. Additionally, we recovered a higher proportion of known ENCODE peaks in our new dataset (up to 63% in a single experiment – for H3K27me3) than the one presented in our previous manuscript (up to 50% in a single experiment). We are confident that this should address the main concerns of the remaining reviewers. For full transparency, we are still including the original data in the supplements, but all results and conclusions regarding computational optimization and benchmarking are now based on the new experiments.

Although the authors did a thorough job of testing different H3K27ac antibodies, now the major antibody companies are doing the same for CUT&RUN/Tag with at least histone modifications as they have in the past for ChIP-seq. As for CUT&Tag recovering at most half of the ChIP-seq peaks, this says more about peak callers than data quality, and who knows what the authors would have found if they had tested ChIP-seq against CUT&Tag peaks? Not that I'm suggesting doing that but rather to use annotated landmarks that are not biased by setting one method as a standard with the goal for the other being to capture all the peaks. As we don't know which of the ENCODE peaks are of functional significance, it seems possible that the ones found by both methods are the ones that users should care about. Instead, the authors could

do their analysis by asking how well H3K27ac captures ENCODE's candidate cis-regulatory elements (cCREs), which is based mostly on DNaseI hypersensitivity data collected by the project. Or they could use annotations based on function, such as STARR-seq (PMID: 33292397) for enhancers and SuRE (PMID: 28024146) for promoters, available for K562 cells.

We agree that the discrepancy between ENCODE ChIP-seq and CUT&Tag is not one-sided, this is the nature of differences and overlaps. As explained above, rather than a ground truth, ChIP-seq and ENCODE are the assay and data reference that have been overwhelmingly used to date for studies of complex disease and therefore make a relevant reference point. Our work raises questions about the quality of ChIP-seq, which we articulate transparently in the manuscript (eg, “it is unclear exactly how well ENCODE data reflects the ground truth. ChIP-seq peaks missed by CUT&Tag could potentially represent noise or false signal detected by ChIP-seq due to chromatin shearing and sonication, as well as fixation and cross-linking resulting in heterochromatin bias.”).

In accordance with published work and Professor Henikoff's FRiP analysis in his review, we show better signal enrichment for H3K27me3 CUT&Tag compared to ENCODE across multiple metrics, including high FRiPs, lower noise signal tracks, higher enrichments of reads around peak summits and a more localized genome-wide signal. This striking enrichment in signal-to-noise is not as evident for H3K27ac CUT&Tag data across all metrics. Shown below are lower fragments in peaks (both ENCODE and those defined by MACS2 or SEACR in our own data), as well as no clearly pronounced advantage in genome-wide signal localization compared to ENCODE H3K27ac:

Professor Henikoff makes good suggestions for orthogonal approaches and data, which could help determine what the ground truth in histone modification signals is. Of note, we have performed overlap analyses with STARR-seq data, which showed the following:

“ENCODE H3K27ac ChIP-seq recovered significantly more STARR-seq peaks than CUT&Tag, even when restricting to STARR-seq peaks overlapping with open chromatin regions that were profiled by DNase-seq experiments. However, when adjusting for genomic coverage overlap was highly comparable with MACS2 CUT&Tag (**Supplementary Fig. 6a**). This was accompanied by a shift towards higher q-value distributions of STARR-seq peaks captured versus missed by CUT&Tag and ENCODE (**Supplementary Fig. 6b**).”

As a proxy for regulatory activity we performed extensive overlap analyses with published ENCODE ATAC-seq data, which confirm that H3K27ac CUT&Tag is most enriched at open chromatin H3K27ac ENCODE peaks.

My FRiP analysis also confirms that the ChIP-seq FRiPs using GSM788088 from ENCODE using CST9733 (purple curves) are much lower than CUT&Tag, both for our 2019 data and for the authors' data when all are down-sampled similarly. Here I'm comparing cell numbers down to 60 cells, and even though ENCODE ChIP-seq used 1 million cells, our low-cell number data are cleaner. So setting ChIP-seq as the standard, and strongly implying with a declarative statement in the title that somehow CUT&Tag doesn't perform as well is misleading and ignores what researchers who are trying to decide which method to use really care about.

Steve Henikoff
 HHMI and Basic Sciences Division
 Fred Hutchinson Cancer Center

Professor Henikoff's experiments on very low cell numbers are hugely impressive and a major advance for the field. However, in this manuscript, we are not trying to establish the boundaries of low input assays, but rather asking the question of how

ChIP-seq and CUT&Tag (particularly for H3K27ac) compare under favourable conditions with plenty of input material and high sequencing depths, as one would expect for most epidemiological studies conducted on tissues and common cell types. Overall, we think our results are very much in line with what Professor Henikoff has shown in the past, indicating a clear improvement of signal to noise in H3K27me3 CUT&Tag over ENCODE, which, however, doesn't seem to apply in the same way to H3K27ac CUT&Tag. We trust that our revised manuscript, and most importantly, our newly generated high-quality, low-duplication CUT&Tag data give the reviewers confidence in our results and address any remaining concerns.